# Hidden impacts of ocean warming and acidification on biological responses of marine animals revealed through meta-analysis

Katharina Alter [1,15] ✉, Juliette Jacquemont [2,3,15], Joachim Claudet [3], María E. Lattuca [4], María E. Barrantes[5], Stefano Marras[6], Patricio H. Manríquez [7,8], Claudio P. González[7,8], Daniel A. Fernández[4,5], Myron A. Peck [1,9], Carlo Cattano [10,11], Marco Milazzo [10,12], Felix C. Mark [13] & Paolo Domenici [6,10,14]

Conflicting results remain on the impacts of climate change on marine organisms, hindering our capacity to predict the future state of marine eco-systems. To account for species-specific responses and for the ambiguous relation of most metrics to fitness, we develop a meta-analytical approach based on the deviation of responses from reference values (absolute change) to complement meta-analyses of directional (relative) changes in responses. Using this approach, we evaluate responses of fish and invertebrates to warming and acidification. We find that climate drivers induce directional changes in calcification, survival, and metabolism, and significant deviations in twice as many biological responses, including physiology, reproduction, behavior, and development. Widespread deviations of responses are detected even under moderate intensity levels of warming and acidification, while directional changes are mostly limited to more severe intensity levels. Because such deviations may result in ecological shifts impacting ecosystem structures and processes, our results suggest that climate change will likely have stronger impacts than those previously predicted based on directional changes alone.

The rapid increase in atmospheric carbon dioxide is changing our climate at a pace never observed before, with consequences on global biodiversity and, ultimately, human well-being[1]. Ocean warming (OW) and ocean acidification (OA), caused by the increased partial pressure of carbon dioxide ($pCO_2$) in seawater, and deoxygenation represent the three greatest climatic threats to marine life[2]. Dramatic effects of these three drivers have already been observed not only at the organism level but also at the scale of entire ecosystems[3]. Examining the impacts of climate change on marine life has been one of the most rapidly growing fields of research[4]. Research shows that OW increases energetic costs and decreases the survival of marine organisms and

that OA impacts invertebrates more than fish through adverse effects on survival, calcification, growth, and development[5–11]. Additional factors such as life-stage, taxa, and acclimation time have been demonstrated to significantly alter the sensitivity of marine organisms to climate change drivers[5,7,8,12–14]. Recently, experimental designs have increased in complexity and realism to account for the interaction of simultaneous climate change drivers, although the combined effect of deoxygenation with OW or OA remains understudied[11,15].

An inherent challenge to the richness of the published literature documenting the effects of climate drivers is to design quantitative syntheses that summarize results while accounting for the diversity of

systems tested. Previous meta-analyses testing effects on similar taxa and biological responses have found varying magnitudes of climate driver effects and even different directions of changes (Fig. 1). While publication biases and decline effects (i.e., the decreasing effect of a driver over time) may contribute to this heterogeneity[16–22], conflicting results also arise from differences in methods used to pool data since meta-analyses have either been performed on metrics individually (e.g., "growth rate", "size", and "weight"[20]), grouped by category (e.g., "growth"[11,14]) or all pooled together (e.g., "overall sensitivity"[13]). While testing effects on categories of biological responses rather than on individual metrics increases the statistical power of meta-analyses, this approach requires to attribute a direction to each metric, i.e., whether an increase of the metric is beneficial or detrimental to fitness, so that metrics of opposite directions (e.g., mortality and survival) do not cancel out when aggregated. However, in most cases, the effect of a metric's increase on fitness remains uncertain or is context-dependent. For example, an increase in boldness is linked to longer exploration periods which might result in more success in foraging for food but also may increase mortality due to increased exposure to

predators[23–25]. Similarly, although increases in respiration rates are generally considered to be positively linked to fitness in meta-analyses, such increases can also indicate higher metabolic needs that come at the expense of growth and reproduction[26,27]. Hence, changes in metrics may result in trade-offs rather than in unequivocal benefits or costs to fitness, and, for most metrics, it remains challenging to confidently determine their relation to fitness.

Many meta-analyses have dealt with the ambiguous relation of metrics to fitness by assuming a positive effect in all cases except when a negative effect on fitness is clearly established (e.g., mortality, shell damage)[5,7,11]. However, this assumption may result in mislabeling the direction of more ambiguous metrics, ultimately leading to meaningful but opposite changes in metrics canceling out when averaged and underestimating climate impacts[28]. This risk is amplified when results are pooled across species, ecosystems, and climates because of the importance of species-specific traits in mediating responses to climate change drivers[13,29] and because benefits provided by traits are context-dependent. Some analyses have taken these specificities into account by summarizing results at the taxa level[7], for given species

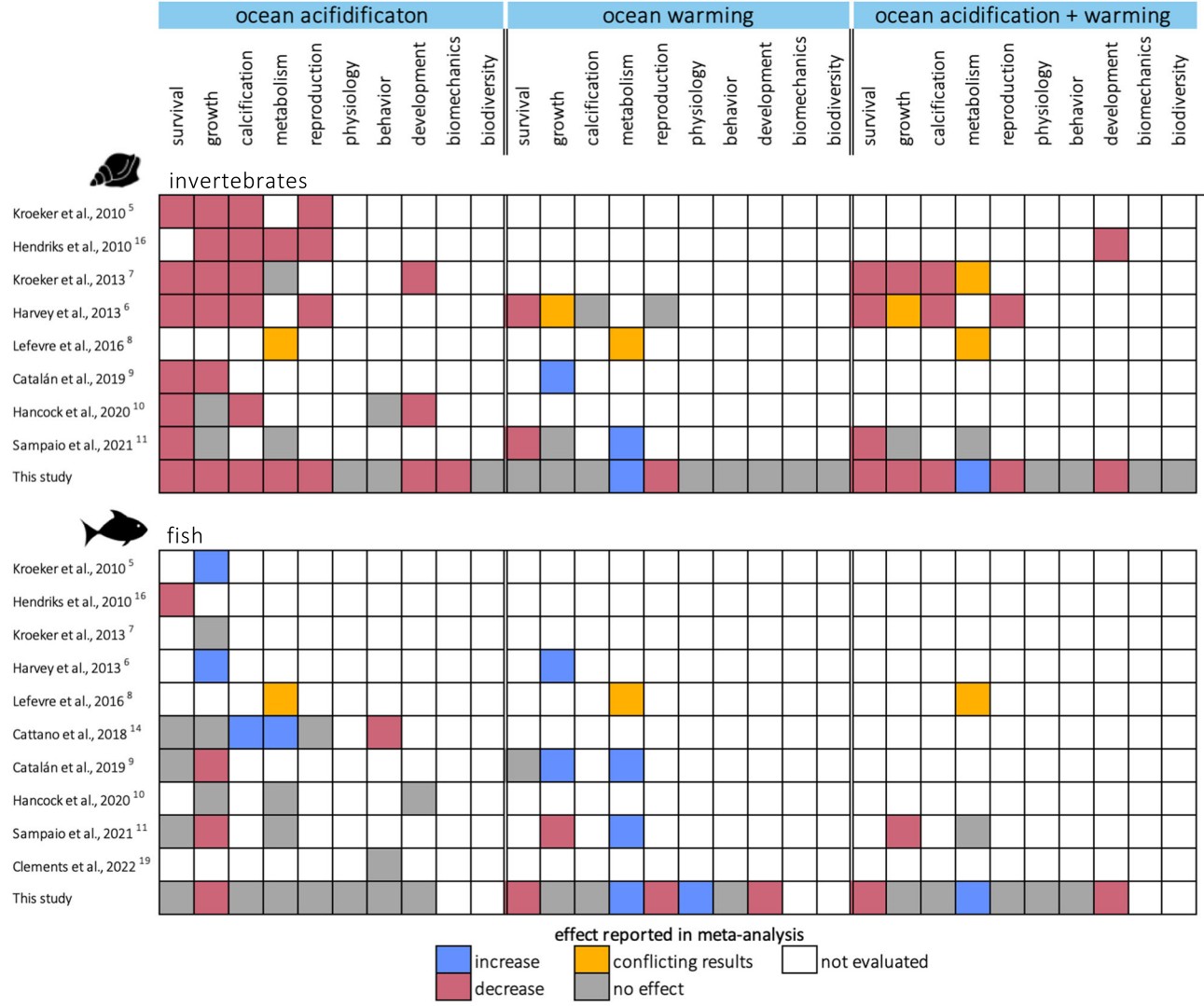

**Fig. 1 | Results of previous meta-analyses on the effects of climate drivers on biological responses of marine animals.** Different colored tiles indicate that a given meta-analysis reported increases (blue), decreases (magenta), conflicting results (i.e., different effects depending on variables tested that were not pooled in the study; orange), no effect (gray) or did not evaluate (white) a given biological response of invertebrates and fish to ocean acidification, ocean warming, and their combination. Data were assessed at the 95% confidence interval level. Fish and mollusc icons are available on the noun project website: https://thenounproject.com/icon/fish-1464319/ and https://thenounproject.com/icon/mollusk-5552214, respectively.

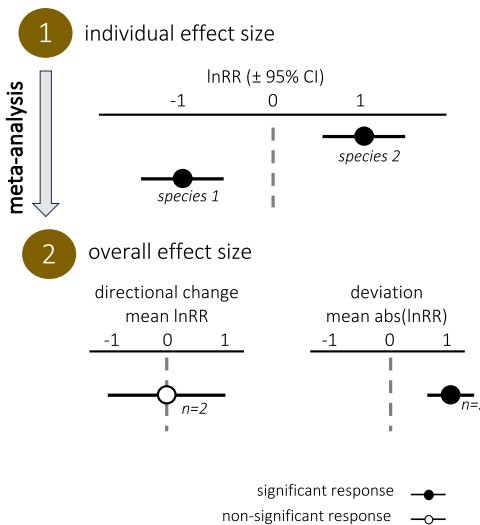

**Fig. 2 | Diagram showcasing differences between directional changes and deviations.** Antagonistic responses at the experiment level can cancel out when computing a mean directional change (lnRR). By contrast, significant responses are revealed when computing mean deviation (abs(lnRR)). CI confidence interval, *n* sample size.

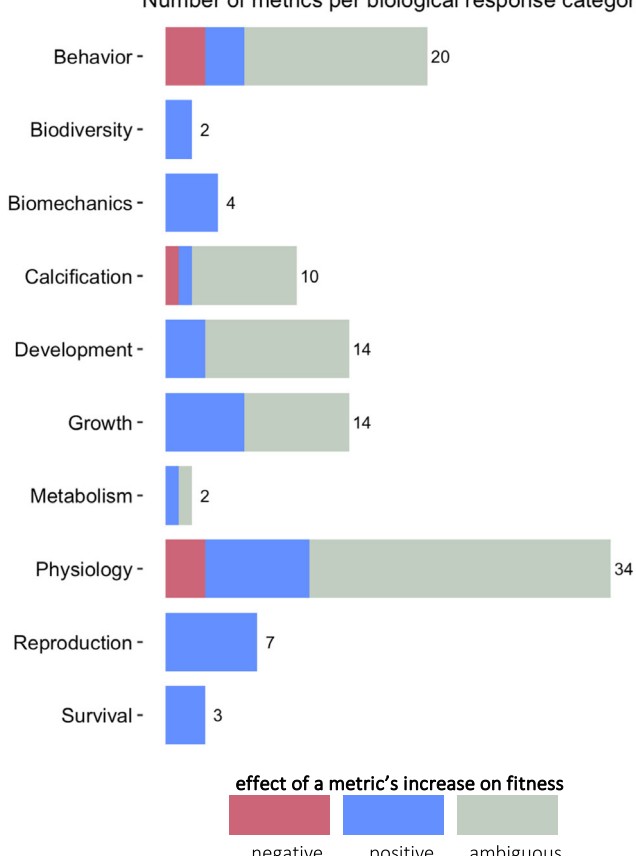

**Fig. 3 | Effect of metric's increase on fitness and number of metrics per biological response category.** Magenta, blue, and gray fillings indicate metrics for which an increase leads to a negative, positive, or ambiguous effect on fitness, respectively. The number of metrics per biological response category included in our analysis is indicated next to each bar. Source data are provided as a Source Data file.

traits[14], life-stages[11], or by presenting individual effect sizes in addition to means[8]. However, disaggregating data comes with the trade-off of lower statistical robustness and generates multiple heterogeneous results that obscure overall trends.

A different approach is therefore needed to overcome our limited understanding of the effect of metrics' changes on fitness and to mitigate the risks of underestimating effects when pooling data. We propose that testing for deviations of biological responses, i.e., absolute distance to reference value, can be used to complement the traditional directional change approach, i.e., relative distance to reference value, to detect impacts of climate drivers on marine life (Fig. 2). We argue that testing for deviations of biological response to climate drivers is meaningful because any significant deviation from the reference state of a metric value, whether "positive" or "negative", can cause cascading changes up to community and ecosystem levels[30–32]. Testing for deviation of metrics rather than for directional changes is a widespread approach in medical fields such as human physiology and cognition[33,34] and has recently been applied to test the effects of environmental drivers on the abundance of fish species[35].

Here, we conduct a meta-analysis testing for deviations in biological responses under climate change drivers to complement the directional meta-analytical approach that has so far dominated this field. We first review metrics measured in the literature and evaluate which ones can confidently be linked to either adverse or positive effects on fitness, which is necessary to interpret results from directional meta-analyses. Then, we test the effects of OW, OA, and their combination on marine organisms by evaluating both directional changes and deviations in ten categories of biological responses. We analyze the impacts of climate drivers for invertebrates and fish separately and for three intensity levels of OW and OA: levels predicted for 2100 under IPCC Representative Concentration Pathways 6.0 and 8.5 (RCP 6 and RCP 8.5), and levels exceeding RCP 8.5 (hereafter "extreme level"). Finally, we compare significant effects detected when testing for directional changes with those detected when testing for deviations. We find significant deviational effects of climate drivers in twice as many biological responses of fish and invertebrates than when testing for directional effects. We detect widespread deviations of responses even under moderate intensity levels (IPCC RCP 6) of OW and OA for 2100, while directional changes are mostly limited to higher intensity levels (RCP 8.5 and extreme). Our results highlight the risks of

underestimating the impacts of climate change on biological response and reveal impacts of climate change that were until now hidden by counterbalancing effects.

## Results and discussion
### Relation of metrics to fitness
We identified 217 studies that investigated the combined effect of OA and OW on marine organisms, yielding 3162 control-treatment comparisons testing different species, climate driver levels, or metrics. We grouped metrics into ten categories of biological responses, and restricted data extraction to two metrics per biological response per study, selecting the metrics most frequently measured in the literature (see Methods and Supplementary Data 2 for details on metric selection). This resulted in the extraction of data documenting 110 metrics, which were evaluated by experts' judgment for their effect on fitness (Supplementary Data 4). Five out of the ten biological response categories included over ten metrics (Fig. 3), with physiology and behavior being measured through the broadest range of metrics (*n* = 34 and *n* = 20, respectively). Only four biological responses (biodiversity, biomechanics, reproduction, and survival) were entirely measured by metrics for which an increase is associated with a non-ambiguous (i.e., positive or negative) effect on fitness (Fig. 3). By contrast, 50 to 80% of metrics used to measure the six other biological responses (behavior, calcification, development, growth, metabolism, and physiology) have an ambiguous relation to fitness either because of lack of knowledge or because of context-dependent effects.

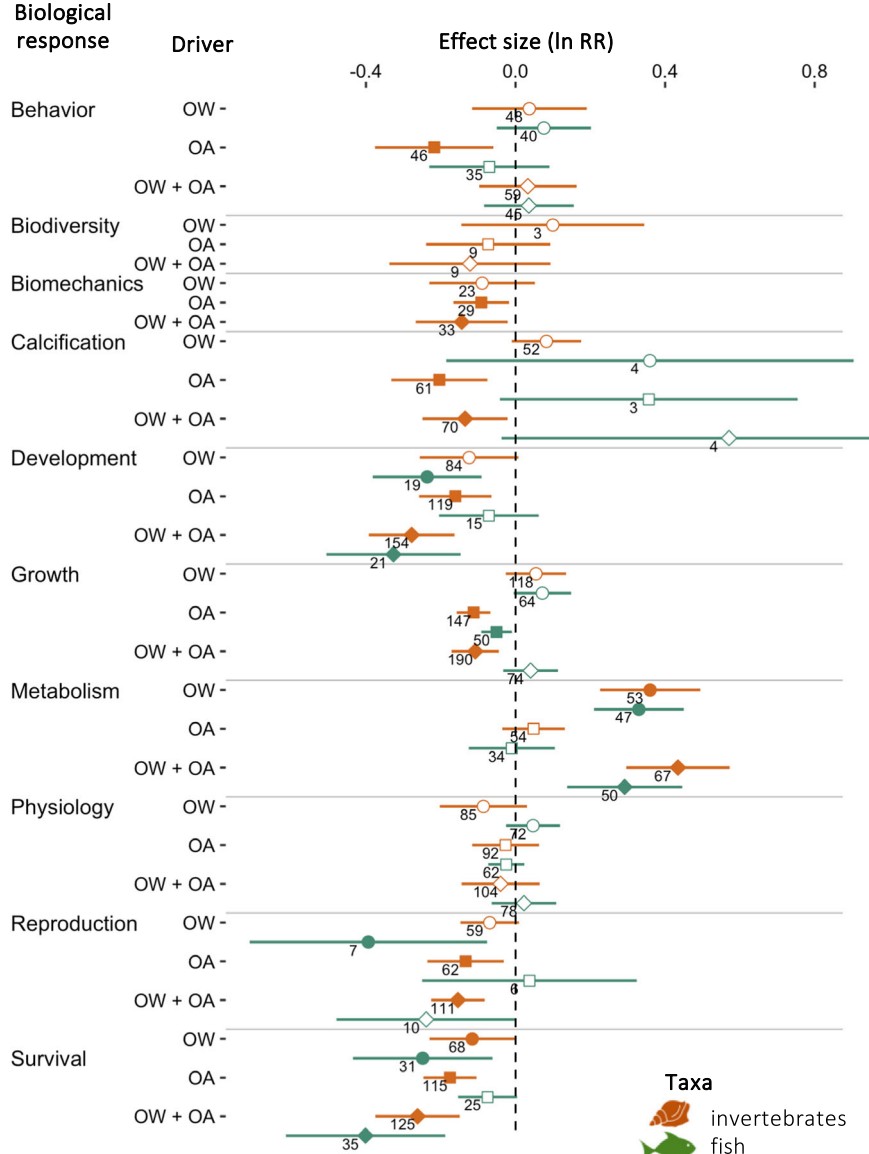

**Fig. 4 | Directional effects of climate drivers on biological responses of marine animals.** Directional effect (lnRR) of ocean warming (OW, circles), ocean acidification (OA, $pCO_2$, squares), and their combination (OW + OA, diamonds) on the biological responses of invertebrates (orange) and fish (green). Significant deviations are denoted by filled symbols (resp., open symbols for non-significant symbols). Error bars represent 95% confidence intervals associated with the mean effect size and numbers indicate sample sizes. Source data are provided as a Source Data file. Fish and mollusc icons are available on the noun project website: https://thenounproject.com/icon/fish-1464319/ and https://thenounproject.com/icon/mollusk-5552214, respectively.

## Directional effects of climate drivers

Following the approach used in previous meta-analyses[5–11,14], we first tested for directional effects of climate drivers on biological responses using logarithm response ratios (lnRR). Due to recent research efforts focusing on previously understudied biological responses, we also synthesized the effect of OA and OW on a community-level response, i.e., invertebrate biodiversity, and increased the number of organism-level responses evaluated for fish and invertebrates in comparison to previous meta-analyses (Fig. 1). However, we did not find any study investigating the effects of combined OA and OW on fish biomechanics or biodiversity. We found that most biological responses (seven out of ten for invertebrates and five out of eight for fish) were significantly affected by OW or OA (Fig. 4 and Supplementary Table 3). OA negatively impacted most biological responses of invertebrates (behavior, biomechanics, calcification, development, growth, reproduction, and survival) but only affected one of eight biological responses of fish (decrease in growth). These results are consistent with previous meta-

analyses[6,7,10,16] and reflect the reliance of invertebrates on the availability of carbonate ions, which decreases under OA[36,37], to build their shells and skeletons[7,38]. By contrast, fish can tolerate higher OA levels than invertebrates[39] due to their elaborate acid-base regulation system[40] and to their bony skeleton composed of calcium phosphate rather than calcium carbonate[41]. We did not find any directional effect of OA on fish behavior, although this could be due to the diversity of fish species pooled or to the diversity of behavioral metrics considered jointly and should be interpreted with caution. The effect of OA on fish behavior is presently a matter of debate[19,21,28,42–44].

OW had more effects on the biological responses of fish than invertebrates (Fig. 4). Stimulation of metabolism and inhibition of survival were observed for both fish and invertebrates, but decreases in development and reproduction were only observed for fish. Larger impacts of OW on fish compared to invertebrates have been hypothesized to derive from greater increases in metabolic costs for this taxa[45]. In comparison to previous meta-analyses, we found similar

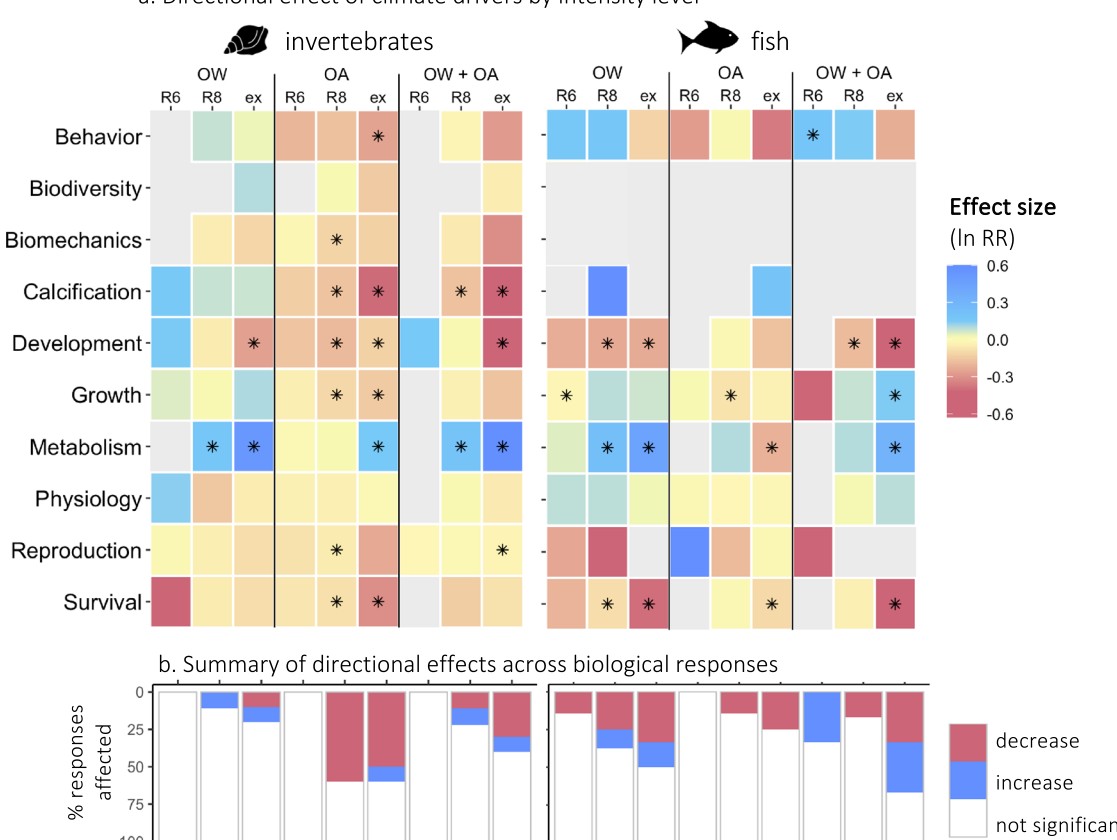

**Fig. 5 | Directional effects of climate drivers by intensity level. a** Directional effects (lnRR) of ocean warming (OW), ocean acidification (OA), and their combination (OW + OA) on biological responses of invertebrates (left) and fish (right) according to the intensity level considered (representative concentration pathway (RCP) 6 (R6), RCP 8.5 (R8), and extreme (ex)). The magnitude of effects is represented by a blue (increase) to magenta (decrease) color scale. Light gray tiles indicate an absence of data. Asterisks indicate significant effects. **b** Proportion of biological responses for which a significant increase (blue) or decrease (magenta) was found for each climate driver and intensity level. Source data are provided as a Source Data file. Fish and mollusc icons are available on the noun project website: https://thenounproject.com/icon/fish-1464319/ and https://thenounproject.com/icon/mollusk-5552214, respectively.

directional effects of responses to OW on invertebrates but fewer effects of OA on fish (Fig. 1). The combination of OW and OA (OW + OA) resulted in fewer and smaller effects than OA alone on invertebrates, or OW alone on fish (Fig. 4). These results suggest antagonistic effects of OW and OA, which support findings from previous studies[7,11], although context-dependent synergistic and additive effects have also been reported[6]. When significant, responses of invertebrates to OW + OA mostly mirrored responses to OA, while for fish, they mostly mirrored responses to OW (Fig. 4), reflecting the climate driver most impactful to these respective taxa.

**Impact of climate driver level**

The number of biological responses affected by OW and OA, as well as the magnitude of these responses, increased with the intensity level of climate drivers (Fig. 5, Supplementary Data 5, Supplementary Table 1). For invertebrates, exposure to RCP 6 levels of climate drivers did not induce any directional response. Yet, under RCP 8.5 levels, OW increased metabolism, OA decreased survival, reproduction, growth, development, calcification, and biomechanics, and their combination increased metabolism and decreased calcification. Similarly, we found only two significant effects of RCP 6 level drivers on fish directional responses: inhibition of growth under OW and increases in behavioral responses under OW + OA. Effects on fish were more pronounced under RCP 8.5 levels: OW decreased survival and development and enhanced metabolism, while OA reduced growth, and their combination inhibited development. More directional responses were affected under extreme levels of drivers (exceeding RCP 8.5) for both

invertebrates and fish (Fig. 5). These trends are consistent with previous results on OW or OA individually[13,14,46] and document, for the first time, this pattern for the combination of these drivers. Currently, RCP 8.5 levels of climate drivers have been tested sixfold more often than RCP 6 levels. The underrepresentation of less severe levels of climate drivers hinders our ability to evaluate the ecological outcomes associated with achieving different RCP trajectories and limits our capacity to predict and manage for near-term impacts of OW and OA. The smaller sample size associated with RCP 6 levels might also contribute to the limited effects detected in this study and calls for further research efforts.

The intensity level of an experiment depends on the choice of its control value, which should account for the mean local environmental conditions but also for the variability and extreme conditions that organisms experience during their development. However, $pCO2$ control values used in studies are sometimes based on $pCO2$ values for the open ocean, which can strongly differ from local coastal $pCO2$ conditions[47,48]. For this reason, it has been suggested to measure intensity levels of experimental OA using a $\Delta pCO2$ exposure index based on local $pCO2$ upper conditions rather than on control values provided by studies[46]. Applying this approach, we found a significant correlation between the $\Delta pCO2$ exposure index and the magnitude of both directional and deviational responses, yet the data fit was similar to that based on $\Delta pCO2$ as provided in studies (Supplementary Fig. 2). Similarly, responses of invertebrates to RCP 6, RCP 8.5, and extreme levels of OA were stable using either study-based or exposure index $\Delta pCO2$, i.e., 75 and 79% of significant responses were shared using both

approaches for directional and deviational effect sizes, respectively (Supplementary Figs. 3, 4). While the exposure index approach is currently restricted to sessile organisms and $p$CO2 treatments, adapting this methodology to accommodate the study of additional climate drivers and their combination, as well as mobile organisms, could provide further insights to elucidate drivers of organisms' response to climate change.

## Deviations of biological responses

Because of the diversity of species, experimental designs, and metrics tested in the literature, and because of the predominance of metrics with ambiguous relation to fitness, we posit that restricting analyses of climate impacts to mean directional changes across studies can be misleading. When pooling different species and metrics, changes of opposite directions can cancel out, masking individually significant changes (Fig. 2). This is problematic as the deviation of any response from its reference state holds biological significance by altering the balance at the cellular, organism, or ecosystem scale. Deviation of responses requires thorough consideration and testing when evaluating climate change impacts and cannot be captured by meta-analyses based on relative effect size. For this reason, we converted relative effect size into absolute effect size ($|\ln RR|$) to calculate the average deviation in biological responses across studies. By mathematical construction, all significant directional changes translate into significant deviations, but significant deviations can be found in the absence of significant directional change, because, unlike relative effect sizes, absolute effect sizes do not cancel out when averaged (Fig. 2).

We found that OW, OA, and their combination caused significant deviations in all biological responses of invertebrates and fish, except for fish calcification and fish reproduction under OA (Fig. 6 and Supplementary Table 4). For a given climate driver, significant effects were detected in up to eight times more responses when testing for absolute deviations than for directional changes (Fig. 7 and Supplementary Data 5 and 6). This was especially true for biological responses described through numerous metrics with ambiguous effects on fitness, such as behavior and physiology (Fig. 3), which supports our hypothesis that antagonistic effects might be hidden when testing for directional changes in such responses. We also found significant deviations in most biological responses of invertebrates under OW and of fish under OA, for which we had detected limited directional changes (Fig. 7 and Supplementary Data 5 and 6). Similarly, we found significant deviations in the behavior and physiology of fish and invertebrates under OW + OA whereas no directional change was detected. This is in line with the finding that a number of behavioral effects in fishes can be mediated by neurophysiological or sensory mechanisms[22], the effects of which may be revealed only when deviations rather than directional changes are taken into account. Moreover, we observed significant deviations in the responses of invertebrates to RCP 6 levels, whereas no directional changes were detected (Supplementary Table 5). In contrast, we found no additional effect of RCP 6 level drivers on fish responses when testing for absolute deviations compared to testing for directional changes. The deviation of responses under OW + OA closely mirrored that of responses under OW or OA alone, depending on taxa and response, and globally followed a trend of antagonistic effect of OW + OA.

Although the challenge of including ambiguous metrics in meta-analyses has been previously recognized[16], the most common approach has been to exclude them from analyses or to assign them a positive direction as a default before pooling them[5–7]. Our results suggest that these approaches underestimate the effects of climate change because ambiguous metrics that are pooled can be antagonistic and cancel out in the overall average effect size. The only alternative approach has been to report metrics separately[20] or to analyze metrics that have opposite directionalities in independent categories (e.g., boldness being assessed separately from other behavioral metrics[14]).

## Effect of life-stage and acclimation time

Organisms' life-stage (embryo, larvae, juvenile, or adult) had a significant effect on responses to climate drivers. In both fish and invertebrates, early-life-stages (embryo, larvae, juveniles) displayed more significant directional responses than adults (Supplementary Figs. 5, 6 and Supplementary Table 2). Early-life-stage invertebrates predominantly displayed significant decreases in responses (Supplementary Fig. 5), while early-life-stage fish displayed both significant increases and decreases (Supplementary Fig. 6). Under OW and OA + OW but not OA alone, biological responses of fish embryos were decreased, and those of juveniles were increased. Biological responses of fish larvae were decreased under OA and increased under OW. For both invertebrates and fish, deviations of responses were significant and similar in magnitude across life-stages and climate drivers, with the exception of embryos' responses that were lower in magnitude. The lower magnitude of deviations, but higher magnitude of directional response of embryos compared to more advanced life-stages, could be due to the less ambiguous and less diverse metrics measured on embryos, typically related to survival and "normality" of developmental processes, leading to fewer counterbalancing effects when computing overall relative effect size. The higher sensitivity of early-life-stages to climate drivers has been found in some, but not all, previous meta-analyses and has been attributed to their lack of regulation and protection mechanisms to cope with environmental changes (ref. 11 vs. ref. 14). Conversely, acclimation time had limited to no effect on directional and deviational responses of organisms (Supplementary Figs. 7, 8). This is consistent with previous meta-analyses that did not find a clear effect of acclimation time on organisms' response[7], and suggest that the influence of acclimation time is likely overshadowed by stronger drivers of responses at the meta-analytical scale, such as life-stage, metric category, or intensity of climate driver level.

## From deviations in the responses of organisms to ecological shifts

The relevance of examining the deviational effects of climate drivers is linked to characteristics of biological processes from the cellular to the ecosystem level. Over evolutionary time scales, organisms have adjusted their metabolic machinery to achieve physiological homeostasis at the lowest metabolic cost possible within the range of conditions of their local environment[49]. Any deviation from an optimal setpoint of homeostasis, whether originating from a metric increase or decrease, is energetically costly as it induces metabolic regulation and, in some cases, compensatory responses[50]. If abiotic conditions vary within the evolutionarily experienced maxima and minima, physiological regulation will ensure homeostasis, yet regulatory metabolic costs will usually rise with increasing deviation from the setpoint[51]. As such, deviation in physiological responses might provide a valuable indicator of the level of stress that organisms are experiencing.

At the population and ecosystem scales, antagonistic responses of different species to climate drivers are unlikely to result in a net absence of change as reflected by directional effect sizes, but rather in a shift of community composition and ecosystem structure[52,53]. Indeed, an increase in the growth rate of a given species will induce cascading effects to predators and prey through trophic interactions, and to competitors because of finite resource availability, resulting in significant shifts in the ecosystem structure. For example, OA has been observed to decrease the relative feeding performance of bivalves and sea urchins in comparison to gastropods[20,54], and to increase the relative growth of turf algae in comparison to kelp[54]. Taken together, these changes induced a shift in the habitat-forming species of this ecosystem from kelp to turf algae[54]. Similarly, studies investigating the effects of climate change on marine biodiversity have reported a reshuffling of species rather than a net loss[53,55]. None of these shifts can be detected at the meta-analytical level by averaging relative distances to reference

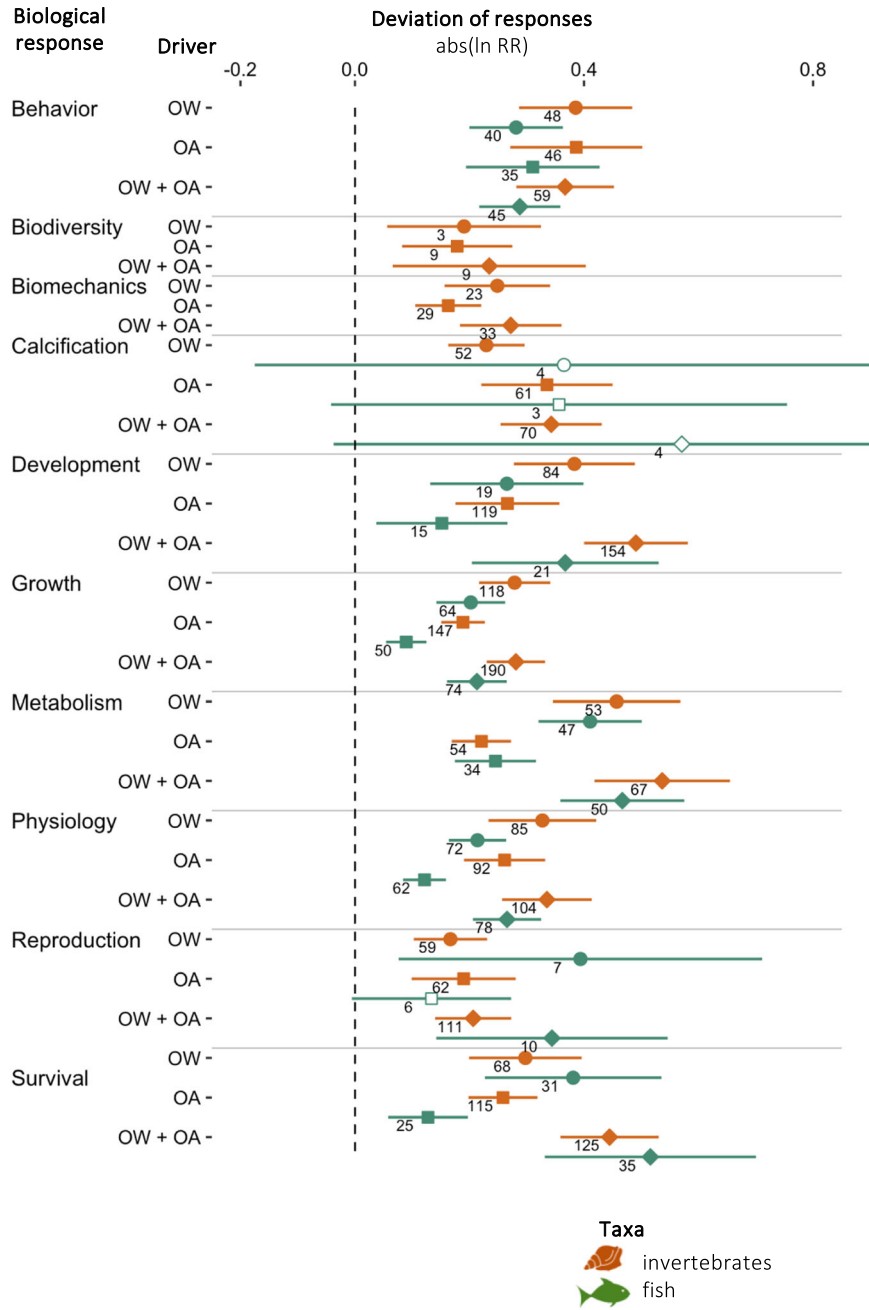

**Fig. 6 | Deviation effects of climate drivers on biological responses of marine animals.** Deviation (abs(lnRR)) of biological responses of invertebrates (orange) and fish (green) to ocean warming (OW, circles), ocean acidification (OA, squares), and their combination (OW + OA, diamonds). Significant deviations are denoted by filled symbols (resp., open symbols for non-significant symbols). Error bars represent 95% confidence intervals associated with the mean effect size and numbers indicate sample sizes. Source data are provided as a Source Data file. Fish and mollusc icons are available on the noun project website: https://thenounproject.com/icon/fish-1464319/ and https://thenounproject.com/icon/mollusk-5552214, respectively.

states but can be detected by averaging absolute distances, i.e., deviations in responses. However, we acknowledge that testing for deviation is less conservative than testing for directional effects because the former decreases the variability of results, thereby increasing the magnitude or significance of climate effects. Therefore, we suggest that deviation and directional analyses should be performed jointly.

**Perspective and future directions**

Although experimental designs have increased in complexity to better reflect real-life systems, several knowledge gaps and limitations remain and hinder our understanding of the current and future impacts of climate change on marine life. Most studies we reviewed relied on short exposure times to test the impacts of climate drivers (51 ± 7 days; mean ± SE), which cannot account for long-term adaptive responses through phenotypic plasticity or adaptation across generations. Conversely, the effects of short, acute climate drivers, such as heat waves or extreme OA events, remain paradoxically understudied in comparison to the gradual effects of OA and OW[11]. This limits our capacity to predict near-term impacts of climate change, characterized by increased frequency and intensity of extreme events and milder average increases in OW, OA, and other climate drivers[56]. This represents a problematic mismatch with the timescale of information needed to inform present-day adaptive management interventions attempting to limit impacts and enhance the resilience of socio-

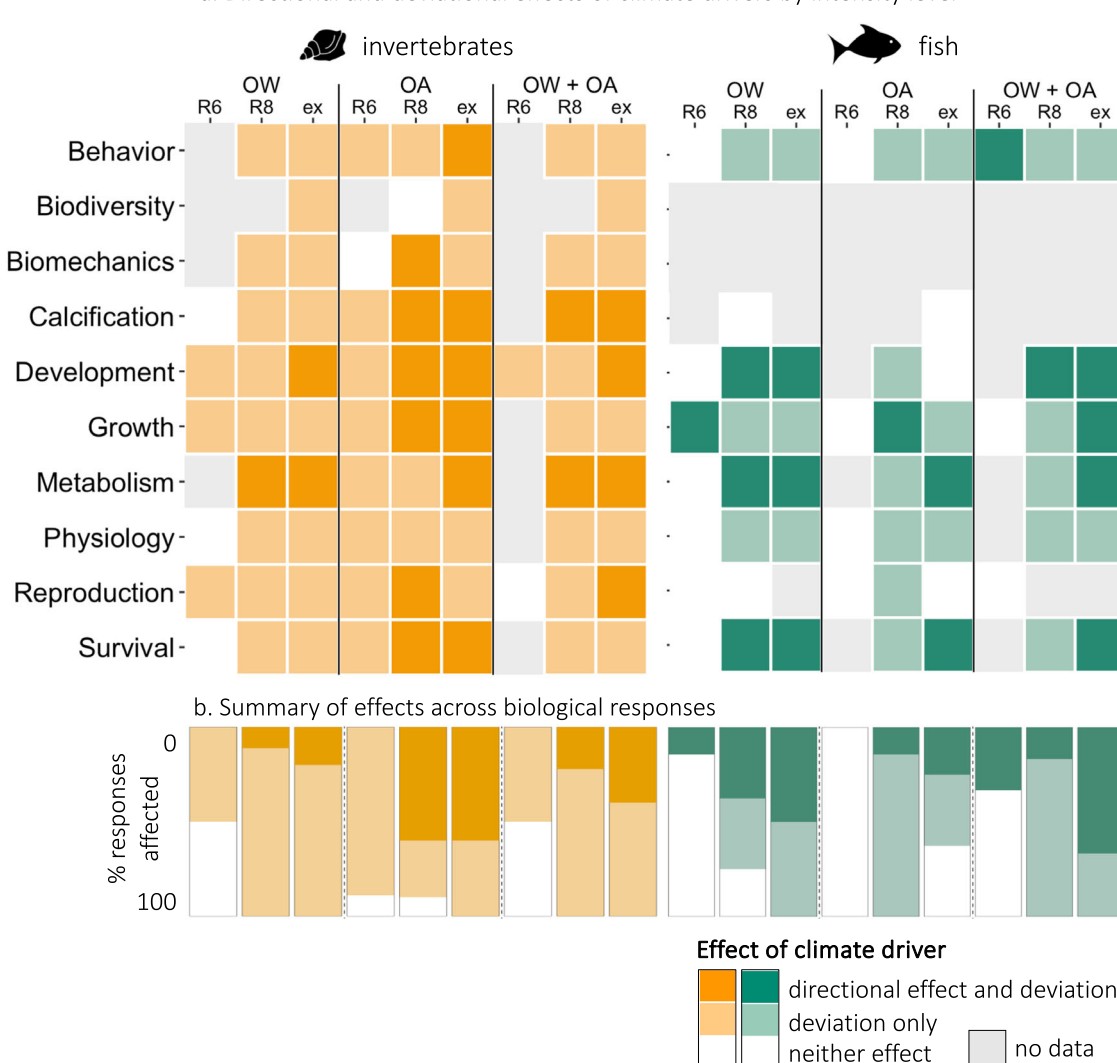

a. Directional and deviational effects of climate drivers by intensity level

b. Summary of effects across biological responses

**Effect of climate driver**

- directional effect and deviation
- deviation only
- neither effect
- no data

**Fig. 7 | Directional and deviational effects of climate drivers. a** Effects of ocean warming (OW), ocean acidification (OA), and their combination (OW + OA), on biological responses of invertebrates (left, orange) and fish (right, green) according to the intensity level considered (representative concentration pathway (RCP) 6 (R6), RCP 8.5 (R8), or extreme (ex)). Colors indicate significant directional and deviational effects (darkest colored tiles), significant deviational effects only (light colored tiles), or no significant effects (white tiles). The absence of data is indicated by gray tiles. **b** Proportion of biological responses (%) for which significant directional and deviational effects (darkest colored tiles), significant deviational effects only (light colored tiles), or no significant effects (white tiles) were found under each intensity level for invertebrates (left, orange) and fish (right, green). Note that all significant directional effects imply significant deviations. Source data are provided as a Source Data file. Fish and mollusc icons are available on the noun project website: https://thenounproject.com/icon/fish-1464319/ and https://thenounproject.com/icon/mollusk-5552214, respectively.

ecological systems within the upcoming decade rather than at the end of the century. Another important consideration is the role played by environmental variability in the adaptive capacity of organisms. Most studies investigating the biological effects of climate drivers have been conducted by exposing organisms to stable experimental conditions even though environmental variability is the norm in nature[46]. While evidence remains scarce, recent studies have shown that organisms tend to be more tolerant of climate change drivers when exposed to fluctuating conditions[47,57].

Finally, more knowledge on the mechanisms that link biological changes to ecosystem structure is needed to predict how deviations or directional changes of responses at the organism level translate into ecosystem-level shifts. This involves conducting experiments investigating multi-species systems and biotic interactions, measuring community-level indicators such as species richness, evenness, functional redundancy or trophic structure, as well as conducting in-situ experiments. While designing empirical studies that test for

community-level shifts is challenging, models can provide valuable insight on this matter (e.g., ecosystem-level impacts from changes in fish boldness[58]).

Our study constitutes an important step forward in documenting the impacts of ocean warming, ocean acidification, and their combination on marine life by assessing the broadest range of biological responses to date and by testing both directional changes and deviations of these responses. We argue that metrics commonly pooled in meta-analyses have predominantly ambiguous or context-dependent effects on fitness, which results in mean effect sizes that are difficult to interpret and that likely underestimate climate impacts. We found that many biological responses that appear unaffected when testing for directional effects are, in fact, significantly deviated from their reference state, suggesting more pervasive effects of climate change than previously thought. While more work is needed to ascertain the impact of deviations in organism-level responses at the ecosystem level, accounting for counterbalancing effects when averaging responses

across metrics and species is a fundamental step toward precautionary assessments of climate change impacts on organisms.

## Methods

### Literature search and data collection

**Systematic literature search strategy.** We performed our systematic literature search on Google Scholar and ISI Web of Science following the PRISMA methodology[59] (Supplementary Fig. 1). The following search string was used for ISI Web of Science: (ocean acidification OR carbon dioxide OR CO2) AND (warming OR temperature) AND (fish OR invertebrate* OR mollusk* OR echinoderm* OR crustacean* OR cnidaria OR bryozoan* OR marine organism*). For Google Scholar, we searched the following combination of words: ocean acidification, carbon dioxide, $CO_2$, warming, temperature, ocean warming, fish, and invertebrate for each year between 2008 and 2022 and limited the search results to 100 per year. All papers published before January 2022 were included in our systematic review. In addition, the reference lists from the retrieved publications, as well as those from previously published meta-analyses on the effects of OA or OW on marine life[11,19], were cross-checked to find publications containing relevant data.

**Screening criteria.** We retained studies that tested the combined effects of OA and OW on marine ectotherms, i.e. fish (teleosts, elasmobranchs) or invertebrates (*Annelida*, *Arthropoda*, bryozoa, *Cnidaria*, *Echinodermata*, *Mollusca*, *Nematoda*, Platyhelminthes, and Porifera). To be considered in the analysis, publications had to include at least two $pCO_2$ and two temperatures in a full factorial design and include information on control and treatment values of $pCO_2$ and temperature. This was done so that antagonistic, synergistic, or additive effects of OA and OW could be evaluated. Only studies that used $CO_2$ or $CO_2$-enriched gas to manipulate $pCO_2$ were kept, those using acid addition were excluded. We counted studies as testing for OA or OW effects if this was the explicit goal of the experiment. For example, we excluded a study that registered an increase of $pCO_2$ of 86 µatm because this was only an undesired parameter change that occurred during a temperature experiment[60]. We excluded all studies that did not report mean values, sample size, or one of the following error types: variance, standard error, standard deviation, or 95% confidence interval.

**Data extraction.** We extracted quantitative data from the text, tables, and graphs of publications using the software GetData, Graph Digitizer, and WebPlotDigitizer. For each study, we extracted information on the biology and ecology of the studied organism (phylum, family, species, life-stage, climatic zone, habitat) as well as information on the experimental design of the study (climate driver tested, climate driver level, and biological metric measured). For each tested driver in a study, we recorded control and treatment values (temperature in °C, $pCO_2$ in µatm), and the associated biological response variables (mean, error, and sample size). Control conditions for $pCO_2$ and temperature were chosen based on the conditions at which the organism was sampled in the wild and acclimated, or in the case of laboratory-raised organisms, the conditions stated in the paper as representing the common biotic range for that organism. Experiments that tested temperature or $pCO_2$ conditions that were lower than the control conditions were not extracted. In the case of studies testing for more factors than OA and OW (e.g., oxygen, salinity, food level), we only extracted data from experiments in which those factors had control values. Data from trans-generational studies were kept only for the parent generation, i.e., the generation that was exposed to control levels of the climate change driver before experiencing OW and OA.

**Classification of metrics.** We grouped metrics among ten biological responses: behavior, biodiversity, biomechanics, calcification,

development, growth, physiology, reproduction, metabolism, and survival (Supplementary Data 3). Then, we attributed a direction (positive, negative, or ambiguous) to each metric according to whether an increase in that metrics' value was considered beneficial, detrimental, or ambiguous (and/or unknown) to the organism, species, or community. The scores were given based on the expertise of five of the co-authors (KA, PD, MM, CC, and FCM), who reviewed metrics and assigned them a direction independently (Supplementary Data 4). We adopted the most conservative approach, i.e., we only assigned a positive or negative direction to a metric if all five co-authors unanimously agreed on that direction. We classified all other metrics as having an ambiguous direction (Supplementary Data 4). When two metrics measuring the same phenomenon were measured in opposite ways across studies (e.g., mortality rate and survival rate, morphological normality (%) and abnormality (%)), we converted all metrics to their positive measurement (e.g., survival rate and morphological normality) to increase the statistical power of our analysis.

**Climate scenarios.** We attributed one of three climate scenarios (RCP 6, RCP 8.5, or extreme) to experiments based on the difference of $pCO_2$ and temperature (T) between control and treatment values. The attribution of climate scenarios followed projections from IPCC 2022[1]: RCP 6 scenario for experiments with $\Delta pCO_2 < 350$ µatm or $\Delta T < 2$ °C; RCP 8.5 scenario for experiments with $350 < \Delta pCO_2 < 750$ µatm or $2$ °C $< \Delta T < 4$ °C, and extreme climate scenario when $\Delta pCO_2 > 750$ µatm or $\Delta T > 4$ °C. For experiments combining $pCO_2$ and temperature treatments, an RCP scenario was only attributed if $\Delta pCO_2$ and $\Delta T$ corresponded to the same scenario. The $\Delta pCO_2$ treatments in the selected studies ranged from 78 µatm[61] to 7894 µatm[62], and the $\Delta T$ ranged from 0.9 °C[63] to 12.8 °C[64].

**Number of data points extracted per study.** Multiple data points were extracted from the same study when they corresponded to different drivers, RCP scenarios, species, life-stages, geographic locations, habitats, or biological responses (e.g., a survival metric and a reproduction metric). For biodiversity metrics, the least taxa-aggregated values were extracted because we considered biodiversity at the community level. When a study reported results on different ontogeny (e.g., number of days since hatching) within one life-stage (e.g., juvenile), data corresponding to the most advanced time point within that life-stage was extracted. When experiments were repeated in summer and winter, we kept the data from the summer experiment as this was the season most commonly investigated. When data were collected from several spawning periods, we kept data from the main spawning event. In all other cases (e.g., several clones measured, different times of the day investigated, several body sizes used), values provided in the article were averaged. The variance of the averaged values $s^2_{\text{aggregated}}$ was calculated using Eq. 1:

$$s^2_{\text{aggregated}} = \frac{1}{k^2} \cdot \sum_i s^2_i \tag{1}$$

where $s_i$ is the variance associated with the averaged value $i$ and $k$ is the total number of values being averaged.

**Number of metrics extracted per study.** When several metrics from the same category were reported in one paper (e.g., the activity of three different enzymes, that all fall within the physiology category), we selected a maximum of two metrics to avoid the over-representation of any given study in our dataset. The selected metrics were chosen based on a priority ranking. First, we assigned priority to metrics classified as positive or negative over those classified as ambiguous. Then we inspected if metrics were correlated and only kept the most inclusive one. For example, if the condition index, shell length, and shell weight were measured, we kept only the

condition index to avoid codependency of metrics within the dataset. When metrics were correlated and equally inclusive, we kept the metric most commonly measured across studies. For example, the activities of the two enzymes superoxide dismutase (SOD) and glutathione *S*-transferase were considered to be correlated because they are both proxies for antioxidant capacity, but SOD was kept because it was measured in more studies. If metrics from the same category were as commonly measured across studies, we chose one randomly. Our choices of selected metrics from studies that reported several metrics from the same category are listed in Supplementary Data 2.

### Data analysis

For each treatment $i$, a relative effect size was calculated as the natural logarithm response ratio of the mean response in treatment $i$ over the mean response in control $i$ (Eq. 2):

$$\ln RR_i = \ln \left( \frac{\bar{x}_{\text{treatment},i}}{\bar{x}_{\text{control},i}} \right) \tag{2}$$

In the case of metrics for which an increase is detrimental to fitness (i.e., negative direction, Supplementary Data 3), the log of the inverse (i.e., control/treatment) was calculated, so that an increase would result in a negative effect size. This formula was also applied in the case of metrics of positive direction but with negative values, because an increase of a negative value corresponds to a negative outcome. If an experiment reported a mean value of zero for its treatment or control, or if an experiment reported values of opposite sign (one positive and one negative) for its control and treatment, the experiment was not included in the analysis because they do not allow to calculate log ratios.

Additionally, for each experiment i, an absolute effect size |ln RR$_i$| was calculated as follow (Eq. 3):

$$\ln RR_i = \left| \ln \left( \frac{\bar{x}_{\text{treatment},i}}{\bar{x}_{\text{control},i}} \right) \right| \tag{3}$$

Variance, standard deviations, and confidence intervals associated with control and treatment mean values were converted into standard errors (SE$_{\text{treatment},i}$ and SE$_{\text{tcontrol},i}$ respectively). The within-experiment variance $v_i$ associated the experiment i was then calculated for both relative and absolute effect size as (Eq. 4):

$$v_i = \frac{SE^2_{\text{treatment},i}}{\bar{x}^2_{\text{treatment},i}} + \frac{SE^2_{\text{control},i}}{\bar{x}^2_{\text{control},i}} \tag{4}$$

Experiments measuring survival, morphological abnormalities, or fertilization success sometimes had null or extremely low within-study variance, e.g., as a result of all individuals surviving. Because the rma() function of the {metafor} package has a within-study variance threshold of 0.0001, we attributed the fixed value of 0.0001 to $n = 24$ experiments (from a total of 3162 experiments) for which variance fell under that threshold. We verified that this did not result in a disproportionate weight given to these data points by checking the weights attributed by models to these studies, as detailed in Supplementary Data 1.

**Random-effect model.** We performed all the parametric data analyses using the {metafor} package[65,66]. We used a weighted random-effects model to quantify the effect of treatments on variables. Effect sizes were weighted, accounting for both the within- and among-study variance components. We conducted a meta-analysis for each combination of taxa (2 levels: invertebrate or fish), climate driver (3 levels: OW, OA, and their combination); climate driver level (RCP 6, RCP 8.5 and extreme); and category of biological response (ten levels, see above), which led to 54 models. Model heterogeneity, residual

heterogeneity, degrees of freedom, and $p$ values associated with the 54 models tested are detailed in Supplementary Data 5, 6 (deviation and directional meta-analyses, respectively). We also carried out meta-analyses across these same categories but grouped climate scenarios together (Supplementary Tables 3, 4). A treatment was considered to have a significant effect on a variable when the 95% confidence interval calculated by the model did not overlap zero.

**Covariates.** The influence of the driver intensity (RCP 6, RCP 8.5, and extreme) on both relative and absolute effect sizes was investigated at the taxa x biological response x driver level when the dataset had $n \geq 10$ data points and featured at least two different scenarios populated by at least two data points. The model heterogeneity and residual heterogeneity associated with these models are shown in Supplementary Tables 1, 5 (directional and deviation meta-analyses, respectively).

The influence of life-stage (embryo, larvae, juveniles, or adults) and acclimation time (number of days of acclimation, square-root transformed) on both relative and absolute effect sizes was investigated at the taxa x driver level. Model heterogeneity, residual heterogeneity, and associated $p$ values are provided in Supplementary Table 2.

**Sensitivity analyses.** To test the robustness of our meta-analysis results, we carried out several sensitivity analyses[18] to detect: (1) the presence of a publication bias and of outliers using visual observation of funnel plots (Supplementary Data 1); (2) the sensitivity of our results to publication bias using the Rosenthal's fail-safe number (N$_{fs}$); (3) whether a different outcome could be obtained when correcting for publication bias using Duval and Tweedie's Trim and Fill test[67,68]. Rosenthal's fail-safe number is an estimation of the number of additional non-significant effect sizes required for a significant meta-analysis result to become non-significant. This allowed us to check the sensitivity of results to uncaptured studies. This risk is estimated to be high if N$_{fs}$ is below 5n + 10, with n the number of data points in the meta-analysis. This was not the case for any of our results (Supplementary Table 6). Duval and Tweedie's Trim and Fill test could only be applied to our relative meta-analyses, which were all found to be robust to potential publication bias under this test (Supplementary Table 6).

Outlying effect sizes were identified through the visual observation of funnel plots (Supplementary Data 1). Additionally, their associated weight was checked using forest plots (Supplementary Data 1) to make sure that no unique value was overwhelmingly influencing the overall effect size[18]. The studies corresponding to outlying points were scrutinized for factors that could explain the extreme values found. Because no flaws or marked differences in the experimental design of these studies were found, no points were excluded from our meta-analyses.

**Effect of upper environmental conditions.** We tested the effect of local upper environmental conditions, as a proxy for local variability, on the biological responses of organisms. We limited this analysis to OA following a detailed methodology developed to test the effects of local $p$CO$_2$ extremes on organisms' responses to OA[46]. This methodology can only be applied under a certain number of conditions, i.e., when studied organisms are sessile or have low-vagility and when $p$CO$_2$ data from sampling sites are available. Furthermore, it has not yet been extended to evaluate the effects of local temperature extremes. This would require a novel approach that takes into account other bioclimatic metrics such as diurnal temperature ranges, isothermality, temperature seasonality and range, microclimate as well as thermal acclimation capacity. In addition, many studies do not report the date of the animal collection, the start date of experiments, and thermal conditions in the laboratory before the commencement of experiments, which would be crucial information for such an approach. This

adds to the difficulties associated with developing this approach, which is outside the scope of this study. We checked studies included in our meta-analysis against the selection criteria given in ref. 46. We retained species selected in ref. 46 and included 24 additional sessile and gregarious or low-vagility benthic species (Supplementary Table 7). Out of the 217 studies used in our meta-analyses, 62 met all selection criteria, including 25 studies that were already included in ref. 46 and 37 additional studies (Supplementary Fig. 9).

Upper environmental conditions at the sampling sites of these 62 studies originated from global database and local buoys deployments, and were extracted from the supplementary information in ref. 46 We then calculated (1) a study-based $\Delta pCO_2$ and (2) a $\Delta pCO_2$ exposure index by calculating the difference between the $pCO_2$ treatment value and (1) the $pCO_2$ control value as given in studies, or (2) upper local environmental conditions, respectively.

We tested the relation between study-based $\Delta pCO_2$ and $\Delta pCO_2$ exposure index and the response of organisms using linear regression models. We attributed climate scenarios to each data point following the same procedure as described in the "Climate scenario" section but using the $\Delta pCO_2$ exposure index instead of the study-based $\Delta pCO_2$. Because studies that met the criteria necessary to calculate a $\Delta pCO_2$ exposure index were much fewer than our initial study pool, we performed tests at the biological response x intensity level regardless of sample sizes. Results from linear regressions are shown in Supplementary Fig. 2, and directional and deviational responses by the biological response and by intensity level using both $\Delta pCO_2$ approaches in Supplementary Figs. 3, 4, respectively. The model heterogeneity and residual heterogeneity associated with these models are shown in Supplementary Table 8.

### Reporting summary

Further information on research design is available in the Nature Portfolio Reporting Summary linked to this article.

## Data availability

The data used and generated in this study have been deposited in the Zenodo database with https://doi.org/10.5281/zenodo.10223034 (https://zenodo.org/records/10223034)[69]. Source data are provided with this paper.

## Code availability

The code used to perform this study are publicly available in the Zenodo database with https://doi.org/10.5281/zenodo.10223034 (https://zenodo.org/records/10223034)[69].

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

## Acknowledgements

This work was supported by the ERANet project CLIMAR "Climate-driven Changes in the Habitat Suitability of Marine Organisms" (grant number BMBF DLR01DN17019) (K.A., M.A.P., P.D., S.M., P.H.M., C.P.G., D.A.F., M.E.B., and M.E.L.), the project FutureMARES "Climate change and Future MARine Ecosystem Services and biodiversity" from the European Union's Horizon 2020 research and innovation program under grant agreement No 869300 (M.A.P. and K.A.), the International $CO_2$ Natural Analogs (ICONA) Network funded by Japan Society for the Promotion of Science

(M.M. and C.C.), the National Research and Development Agency (ANID, Chile) (P.H.M.), the National Fund for Scientific and Technological Development (FONDECYT, Grants 1130839 and 1181609) (P.H.M.), the National Scientific and Technical Research Council (CONICET, Argentina, grant numbers: PIP 2934 and PUE 2016—CADIC) (D.A.F. and M.E.L.), the project OCAH-Beagle "Ocean acidification and Hypoxia impacts on high latitude marine coastal ecosystems: the case of the Beagle Channel (Southern Patagonia—Argentina, Chile) from the Prince Albert II of Monaco Foundation under the financing agreement No. 2863 (M.E.L.), Biodivera (MOVE and METRODIVER) (J.C.) and Fondation de France (MultiNet) (J.C.). Fish and mollusk icons were created by Lars Meierto-berens and Qolbin Saliim, respectively, from the Noun Project (CC BY 3.0).

## Author contributions

P.D. initiated the study. K.A. led the data compilation to which P.H.M., C.P.G., M.E.L., M.E.B., S.M., D.A.F., M.M., and C.C. contributed. K.A., M.M., C.C., F.C.M., and P.D. assigned directions to metrics. J.J. performed with the assistance of J.C. the meta-analyses. K.A., J.J., P.H.M., F.C.M., J.C., and P.D. interpreted and discussed the results. K.A. and J.J. wrote the first draft with the support of P.H.M., F.C.M., M.A.P., and P.D. All co-authors contributed to the final version.

## Competing interests

The authors declare no competing interests.

## Additional information

[1]Royal Netherlands Institute for Sea Research, Department of Coastal Systems, P.O. Box 59, 1790 AB Den Burg, The Netherlands. [2]School of Aquatic and Fishery Sciences, University of Washington, 1122 NE Boat St, 98195 Seattle, WA, USA. [3]National Center for Scientific Research, PSL Université Paris, CRIOBE, CNRS-EPHE-UPVD, Maison de l'Océan, 195 rue Saint-Jacques, 75005 Paris, France. [4]Centro Austral de Investigaciones Científicas (CADIC-CONICET), Bernardo Houssay 200, V9410CAB Ushuaia, Argentina. [5]Universidad Nacional de Tierra del Fuego, Antártida e Islas del Atlántico Sur; Instituto de Ciencias Polares, Ambiente y Recursos Naturales (UNTDF - ICPA), Fuegia Basket 251, V9410BXE Ushuaia, Argentina. [6]CNR-IAS, Consiglio Nazionale delle Ricerche, Instituto per lo studio degli Impatti Antropici e Sostenibilità in ambiente marino. Località Sa Mardini, 09170 Torregrande, Oristano, Italy. [7]Centro de Estudios Avanzados en Zonas Áridas (CEAZA), Coquimbo, Chile. [8]Laboratorio de Ecología y Conducta de la Ontogenia Temprana (LECOT), Coquimbo, Chile. [9]Wageningen University, Department of Animal Sciences, Marine Animal Ecology Group, De Elst 1, 6708 WD Wageningen, The Netherlands. [10]NBFC, National Biodiversity Future Center, Palermo, Italy. [11]Department of Integrative Marine Ecology, Stazione Zoologica Anton Dohrn (SZN), Lungomare Cristoforo Colombo, I-90149 Palermo, Italy. [12]Dipartimento di Scienze della Terra e del Mare (DiSTeM), Università di Palermo, Via Archirafi 20, I-90123 Palermo, Italy. [13]Section of Integrative Ecophysiology, Alfred Wegener Institute Helmholtz Centre for Polar and Marine Research, Am Handelshafen 12, Bremerhaven 27570, Germany. [14]CNR-IBF, Area di Ricerca San Cataldo, Via G. Moruzzi N°1, 56124 Pisa, Italy. [15]These authors contributed equally: Katharina Alter, Juliette Jacquemont. ✉e-mail: katharina.alter@nioz.nl

