## [Peer Review File · Nature Communications]

Hidden impacts of ocean warming and acidification on biological responses of marine animals revealed through meta-analysisREVIEWER COMMENTS

Reviewer #1 (Remarks to the Author):

In this study, the authors compared two meta-analytical approaches to assess how (directional, the commonly used effect size) and if (deviation, novel alternative effect size) predicted temperature and CO₂ elevations, alone and in combination, affect marine organisms. The directional effects were assessed by the relative effect size, the log response ratio (lnRR), while the deviation was simply the absolute value of the relative effect size. I.e. for the deviation, the magnitudes are the same as the directional effect size for a given study, but all values are positive. This means that in cases where some studies find an increase in a metric, and other studies find a decrease, the directional effect size might not be significant, because different studies cancel each other out, while the “deviation” effect size could be significantly different from zero, because all the decreases are now counting as increases as well. The end result is that, if one asks IF predicted elevated temperature and CO₂ have an effect on marine organisms, the answer is a solid yes when looking at deviations (i.e. the absolute value of the relative effect size), but conflicting when looking at directional changes. On the other hand, neither approach is very good at summarizing generally HOW marine organisms are affected, because responses may go in different directions for different species/studies, when using the directional effect size, and because any information about direction is lost when looking just at the magnitude and not considering direction, when using deviation. For example, while it is difficult to say whether an increase in metabolism is positive or negative for fitness per se, it certainly has an impact on energy demand and potentially for models that incorporates or model energy demand?

Several meta-analyses have now been conducted on the same topic, but obviously, since new primary studies are coming out all the time, up-to-date meta-analyses do hold value, in my opinion, especially one as thoroughly conducted as this one. The present study points out some important challenges when it comes to interpretation of effect sizes, notable what decreases and increases means for fitness, and the risk of underestimating whether there are effects, when summarizing across conflicting results. While the approach as mentioned may have limited use for more mechanistic research (HOW), there is certainly high value in being able to give a more straightforward answer to policymakers and governments (IF).

Overall, I find the study well worth of publication. The manuscript is well prepared and the methodology appears solid, and I mainly have some comments related to clarity (it is difficult to point this out specifically because line numbers are missing, but hopefully the authors will be able to find the places I refer to, and I have also uploaded an annotated file where I marked the places with yellow).

1) Abstract l. 8: here the authors introduce the “new” effect size, i.e. the deviation. To me, it did not make sense what it was until I was well into the manuscript. What does “deviations of responses” actually mean? The term “deviation” suggest that it is something much more different than what it is: the relative effect size and the absolute value of the relative effect size. I realise that the authors may be pressed for space in the abstract, but I think it would be worthwhile adding a sentence for each approach to explain what they actually are (e.g. using the terms used in the beginning of the Main, absolute and relative distance to reference value?). They are in essence one well-known effect size and one derived from it.

2) “Main” second paragraph l. 13-14: this issue regarding interpreting effect sizes in relation to their effect on fitness is important and also seems an important argument for not looking at directional changes alone. Could this be mentioned in the abstract to frame the study? Also, I think it could be discussed a bit more that mean effect size across studies on different species are problematic, because of the risk of getting effect sizes that are not significant, because different studies cancel each other out. Considering the chosen approach and the findings, this seems like an essential issue, and should be explained. It is currently mentioned in the last sentence of the conclusion, as if it is a finding of the study, but this issue was considered and taken into account already by Lefevre 2016, who because of this possible issue showed individual effect sizes and not just mean effects (as done by most studies). But in the present study Lefevre 2016 has simply been labelled as “conflicting results” in Table 1, while it, though limited to metabolism, could be used to illustrate the exact problem that the present study is trying to solve. Had Lefevre 2016 opted to actually report mean effect sizes, the study would most likely have been labelled “no effect”. A sentence from the paper: “A meta-analysis in the strictest sense has as output a single mean effect size for a given variable (Harvey et al., 2013), or even combining several variables (Przeslawski et al., 2015). This mean effect size can then be tested statistically against a

hypothetical value (commonly zero), or between different animal groups and life stages. Although this approach is attractive because it gives straightforward 'yes or no' answers, it also introduces the risk that potentially interesting patterns and range of responses are overlooked. Given that many species are studied (unlike the situation in medical meta-analyses), the breadth of responses may reflect biological diversity. I therefore chose to examine the diversity of responses by presenting data graphically, rather than focusing only on mean effect sizes."

3) Table 1: there is something weird with the space between the references (second line of one reference is close to first line of the next, rather than its own first line...).

4) "Relation of metrics to fitness" first paragraph l. 4-5 from bottom: here, I think the authors need to specify that they mean increases in a metric (as described in the methods). The metric itself is not something that has an effect on fitness, it is the decrease or increase in a metric that can have an effect. This is especially important when turning to Figure 1 – here, the authors talk about ambiguous, positive and negative effects, but this again would have to be with reference to a certain direction of the change of a metric? The "direction of metric" is a reference to the effect of a change in the metric on fitness, and not whether the metric increases or decreases? To be more clear, and again based on what is written in the methods, would it be more specific to label the categories "Effect of increase in metric on fitness"? Or at least explain more clearly in caption what is meant.

5) Figure 3: In the legend "summary of effects" would it not be more specific to use decrease or increase? Since otherwise it may be confused with effects on fitness, but I assume this is not what is meant as earlier in the manuscript it is indicated that it was not possible to ascertain for many metrics whether an effect is positive or negative for fitness.

6) "Deviations of biological responses" first paragraph l. 8: the authors write that "by mathematical construction ... significant deviations can be found in the absence of significant directional changes", and I completely agree. But maybe a short sentence explaining exactly why this is so would be useful for many readers not familiar in depth with the nature of response ratios and the problem that directional mean effect sizes can

effectively “hide” both positive and negative effect sizes, cancelling each other out when the mean is taken. In fact, if this issue was introduced more clearly earlier in the manuscript (i.e. in the “Main” section), the brilliance of looking at the absolute value would be even clearer.

7) “Conclusion” l. 4-6: the authors write that they “demonstrate that metrics commonly pooled in meta-analyses have predominantly ambiguous or context-dependent effects on fitness”. And while I certainly agree with the essence of the statement, it is unclear how exactly the authors “demonstrated” it. There is no detailed discussion or showing of data; the co-authors individually assessed the metrics, but there is no description of how the co-authors (= experts) argued for and against? There is no data or scores in Table S1, just the final classifications. Rather than “demonstrate” I would therefore use the word “argue”.

8) “Literature search and data collection” second paragraph l. 4: It is specified exactly why it was necessary to restrict studies to those having done temperature and acidification in a full-factorial design, but I assume it has to do with how the effect sizes are interpreted, i.e. in terms of assessing whether they are additive/antagonistic? That would not be possible unless exactly the same studies are used, so maybe this could be clarified (if that is indeed the argument).

9) “Literature search and data collection” fourth paragraph l. 4: here it is explained that it is an INCREASE in a metric that is used as a basis to assess the effect on fitness, but this is not mentioned anywhere earlier, that I can see. But as I have commented on above, it gets confusing when it has not been mentioned.

10) “Data analysis” first paragraph l. 3: Again, there is confusion about what “direction” means. With “negative direction” I assume you mean “For metrics where an increase has a negative effect on fitness”, maybe better to specify more clearly?

11) “Data analysis” first paragraph l. 1-2 from bottom: The first things that strikes me is how weird it is that there were studies with zero variance. Not just small, but actually zero? Was it given as zero or was it not possible to assess from graphs? Next, “the value of 0.0001” was

assigned". Why this exact value? Why not 0.001 or 0.00001? While I do not know how many studies had this issue (maybe this could be mentioned?), and how big an impact it has for the overall results, I think it deserves a bit more explanation/justification.

[**Editorial Note:** Reviewer name has been redacted as they do not wish to be named.]

Reviewer #2 (Remarks to the Author):

This manuscript aims at re-evaluating the literature on both ocean warming, acidification and their combined effects. The general idea to deviate from the classic approach attributing a direction to metrics used to measure the biological impact is original and there is some interesting technical suggestions on how to focus on absolute deviation from a reference value. While the manuscript is well-written, presented, and the data well analysed, there is a strong methodological flaw that compromises the selection of studies, the results, and the interpretation.

One critical factor explaining species sensitivity to temperature or the carbonate chemistry is local adaptation to the present natural variability. The variability is huge between regions but also at a given sampling site, especially on the coastal zone and in temperate regions. The importance of incorporating present natural variability has been recently illustrated by Vargas et al. (2017, 2022) for ocean acidification. While authors are citing these two papers to illustrate the importance of natural variability, they do not consider the main point that biological response is correlated to the deviation from the present range of natural variability in pCO₂/pH. Vargas et al. (2022) also makes the point that a majority of authors are using appropriate controls and treatments in their experimental studies. To resolve this issue, they propose an index that compare the high pH level to the extreme of the present range (instead of the "control" as defined by authors). A true OA and OW treatment should consider the present range of natural variability and chose scenarios that deviates from this range. Many authors are justifying their treatments using open ocean scenarios or neglect variability. When accounting for this, a lot of the apparent contradictions between studies disappeared. Another important point related to the natural variability is that for most metrics, there is not 1 control but a range of conditions (e.g. metabolism) that are within the present range of variability that changes without consequences on fitness.

To truly evaluate the effect of OA and OW (and their combination, which would require a stronger consideration of what is a relevant combination of scenarios), the treatments considered by the authors of the selected manuscript should be critically evaluated in the light of the present natural range of variability at the sampling site. A majority of authors were considering the wrong scenario for their “control” and “future treatment”. Ignoring this lead to a mix of data on plastic response to natural variability with true response to OA and OW and compromise the ability to perform a meaningful meta-analysis.

Reviewer #3 (Remarks to the Author):

The current manuscript provides a new approach to considering analyses of data sets on the impacts of climate warming and seawater acidification on marine species. In addition to the more traditional approach of providing a direction of change and assigning impact based on the change (i.e. positive or negative or unknown), this approach also considers deviations from control in response to some of the physical drivers of change in the ocean for invertebrates and fishes.

As a physiologist, I appreciated this approach as for many of the measurements the community has made in terms of responses to climate change, a response in a particular direction does not allow us to differentiate a positive response to a stressor to restore homeostasis vs. a negative long term response where performance is chronically impaired and reduced fitness would be a consequence. This is critical to accurately predict impacts of climate change. This deviations approach, combined with a directional approach is a good step in the right direction.

I appreciate the amount of work that goes into meta-analyses as extensive as this one. My question though is does it go far enough to provide significant changes in our predictive power on the biological responses to warming and acidification. There is still a fair amount of lumping of data sets, as the authors admit (Lines 319-329). With the focus on 4 factors, perhaps a couple of the most important for understanding vulnerability (life stage) and experimental conditions that allow you to better understand whether the stress response is beneficial in the long term or not (acclimation time) are not considered. I am not sure how

your data is coded but is this some additional analyses that could be run? Understanding which life stage is most vulnerable to deviations will allow for a better prediction of community effects since many fishes and inverts have bipartite life cycles and therefore their "communities" differ.

Lastly, there are a number of environmental physiology experts in this author list. I know the goal is to go beyond single species studies and extrapolate to community level impacts but I would have appreciated a physiologist view of why deviations from controls (away from homeostatic set points) in either direction constrain energy budgets and can lead to fitness consequences if homeostasis is not restored.

ANSWERS TO REVIEWER COMMENTS

Reviewer #1 (Remarks to the Author):

In this study, the authors compared two meta-analytical approaches to assess how (directional, the commonly used effect size) and if (deviation, novel alternative effect size) predicted temperature and CO₂ elevations, alone and in combination, affect marine organisms. The directional effects were assessed by the relative effect size, the log response ratio (lnRR), while the deviation was simply the absolute value of the relative effect size. I.e. for the deviation, the magnitudes are the same as the directional effect size for a given study, but all values are positive. This means that in cases where some studies find an increase in a metric, and other studies find a decrease, the directional effect size might not be significant, because different studies cancel each other out, while the “deviation” effect size could be significantly different from zero, because all the decreases are now counting as increases as well. The end result is that, if one asks IF predicted elevated temperature and CO₂ have an effect on marine organisms, the answer is a solid yes when looking at deviations (i.e. the absolute value of the relative effect size), but conflicting when looking at directional changes. On the other hand, neither approach is very good at summarizing generally HOW marine organisms are affected, because responses may go in different directions for different species/studies, when using the directional effect size, and because any information about direction is lost when looking just at the magnitude and not considering direction, when using deviation. For example, while it is difficult to say whether an increase in metabolism is positive or negative for fitness per se, it certainly has an impact on energy demand and potentially for models that incorporate or model energy demand.

Several meta-analyses have now been conducted on the same topic, but obviously, since new primary studies are coming out all the time, up-to-date meta-analyses do hold value, in my opinion, especially one as thoroughly conducted as this one. The present study points out some important challenges when it comes to interpretation of effect sizes, notably what decreases and increases means for fitness, and the risk of underestimating whether there are effects, when summarizing across conflicting results. While the approach as mentioned may have limited use for more mechanistic research (HOW), there is certainly high value in being able to give a more straightforward answer to policymakers and governments (IF).

Overall, I find the study well worth the publication. The manuscript is well prepared and the methodology appears solid, and I mainly have some comments related to clarity (it is difficult to point this out specifically because line numbers are missing, but hopefully the authors will be able to find the places I refer to, and I have also uploaded an annotated file where I marked the places with yellow).

Reply: We thank the reviewer very much for their appreciation of our work. We added line numbers to facilitate the review process. In our replies below, we refer to the lines in the document showing the tracked changes.

R1.C1: Abstract l. 8: here the authors introduce the “new” effect size, i.e. the deviation. To me, it did not make sense what it was until I was well into the manuscript. What does “deviations of responses” actually mean? The term “deviation” suggests that it is something much more different than what it is: the relative effect size and the absolute value of the relative effect size. I realise that the authors may be pressed for space in the abstract, but I think it would be worthwhile adding a sentence for each approach to explain what they actually are (e.g. using the terms used in the beginning of the

Main, absolute and relative distance to reference value?). They are in essence one well-known effect size and one derived from it.

Reply to R1.C1: We thank the reviewer for this comment. We modified the abstract accordingly and the sentence starting in line 40 now reads: “To account for species-specific responses and for the ambiguous relation of most metrics to fitness, we developed a meta-analytical approach based on the deviation of responses from reference values (absolute changes) to complement meta-analyses of directional (relative) changes in responses.”

R1.C2: “Main” second paragraph l. 13-14: this issue regarding interpreting effect sizes in relation to their effect on fitness is important and also seems an important argument for not looking at directional changes alone. Could this be mentioned in the abstract to frame the study? Also, I think it could be discussed a bit more that mean effect size across studies on different species are problematic, because of the risk of getting effect sizes that are not significant, because different studies cancel each other out. Considering the chosen approach and the findings, this seems like an essential issue, and should be explained. It is currently mentioned in the last sentence of the conclusion, as if it is a finding of the study, but this issue was considered and taken into account already by Lefevre 2016, who because of this possible issue showed individual effect sizes and not just mean effects (as done by most studies). But in the present study Lefevre 2016 has simply been labelled as “conflicting results” in Table 1, while it, though limited to metabolism, could be used to illustrate the exact problem that the present study is trying to solve. Had Lefevre 2016 opted to actually report mean effect sizes, the study would most likely have been labelled “no effect”. A sentence from the paper: “A meta-analysis in the strictest sense has as output a single mean effect size for a given variable (Harvey et al., 2013), or even combining several variables (Przeslawski et al., 2015). This mean effect size can then be tested statistically against a hypothetical value (commonly zero), or between different animal groups and life stages. Although this approach is attractive because it gives straightforward ‘yes or no’ answers, it also introduces the risk that potentially interesting patterns and range of responses are overlooked. Given that many species are studied (unlike the situation in medical meta-analyses), the breadth of responses may reflect biological diversity. I therefore chose to examine the diversity of responses by presenting data graphically, rather than focusing only on mean effect sizes.”

Reply to R1.C2: We thank the reviewer for pointing this out. By conflicting results, we meant that the effects of ocean acidification or ocean warming on a given biological response category was evaluated for different variables, for which different effects were found, and these variables were never pooled together in the study so that a single effect (increase/decrease/no effect) could not be summarized for the study in question. We added this explanation in Table 1 caption: “... conflicting results (i.e., different effects depending on variables tested that were not pooled in the study)...” to clarify this. In addition, we addressed the comment of the reviewer throughout the manuscript to better frame the study:

- 1) In the abstract, we modified the text to include the idea that effect sizes are difficult to interpret in relation to fitness. Please see the changes in **Reply to R1.C1**.
- 2) In the Main/Introduction text, we added an acknowledgement of the approach adopted by Lefevre 2016 to account for potential effect size cancelling out by presenting individual effect sizes. Lines 102 to 107 now read: “This risk is amplified when results are pooled across species, ecosystems, and climates because of the importance of species-specific traits in mediating responses to climate change drivers^{13,29} and because benefits provided by traits are context-

dependent. Some analyses have taken these specificities into account by summarizing results at the taxa level⁷, for given species traits¹⁴, life-stages¹¹, or by presenting individual effect sizes in addition to means⁸.”

In additional sections of the manuscript, we explain why mean effect size across studies on different species are problematic:

- 1) In the Introduction, we added a diagram (Fig. 1) showcasing the differences between directional changes and deviations and how antagonistic responses at the experiment level can cancel out when computing a mean directional change, while significant responses are revealed when computing mean deviation.
- 2) In the results section paragraph “Deviations of biological responses”, we added explanations in the text in lines 229-240: “When pooling different species and metrics, changes of opposite direction can cancel out, masking individually significant changes (Fig. 1). This is problematic as the deviation of any response from its reference state holds biological significance by altering the balance at the cellular, organism, or ecosystem scale. Deviation of responses requires thorough consideration and testing when evaluating climate change impacts and cannot be captured by meta-analyses based on relative effect size. For this reason, we converted relative effect size into absolute effect size ($|\ln RR|$) to calculate the average deviation in biological responses across studies. By mathematical construction, all significant directional changes translate into significant deviations, but significant deviations can be found in the absence of significant directional change, because unlike relative effect sizes, absolute effect sizes do not cancel out when averaged (Fig. 1).”
- 3) The second results section paragraph “From deviation in the responses of organisms to ecological shifts” starts with an explanation in line 299: “At the population and ecosystem scales, antagonistic responses of different species to climate drivers are unlikely to result in a net absence of change as reflected by directional effect sizes, but rather in a shift of community composition and ecosystem structure⁵²⁻⁵³.”

R1.C3: Table 1: there is something weird with the space between the references (the second line of one reference is close to the first line of the next, rather than its own first line...).

Reply to R1.C3: We fixed this formatting problem.

R1.C4: “Relation of metrics to fitness” first paragraph l. 4-5 from bottom: here, I think the authors need to specify that they mean increases in a metric (as described in the methods). The metric itself is not something that has an effect on fitness, it is the decrease or increase in a metric that can have an effect. This is especially important when turning to Figure 1 – here, the authors talk about ambiguous, positive and negative effects, but this again would have to be with reference to a certain direction of the change of a metric? The “direction of metric” is a reference to the effect of a change in the metric on fitness, and not whether the metric increases or decreases? To be more clear, and again based on what is written in the methods, would it be more specific to label the categories “Effect of increase in metric on fitness”? Or at least explain more clearly in caption what is meant.

Reply to R1.C4: We agree that this phrasing could lead to confusion and follow the suggestion of the reviewer. Now we use “effect of metrics’ increase on fitness”. This change was performed throughout the manuscript:

- 1) In the second paragraph of the introduction, lines 89-90: “However, in most cases, the effect of metric’s increase on fitness remains uncertain or is context-dependent”. Further, in lines 95-97: “Hence, changes in metrics may result in trade-offs rather than in unequivocal benefits or costs to fitness and, for most metrics, it remains challenging to confidently determine their relation to fitness.”
- 2) In the “Relation of metrics to fitness” paragraph, line 150 to 155 now read: “Only four biological responses (biodiversity, biomechanics, reproduction, and survival) were entirely measured by metrics for which an increase is associated with a non-ambiguous (i.e., positive or negative) effect on fitness (Fig. 2). By contrast, 50 to 80% of metrics used to measure the six other biological responses (behavior, calcification, development, growth, metabolism, physiology) have an ambiguous relation to fitness either because of lack of knowledge or because of context-dependent effects.”
- 3) In the former Figure 1, now Figure 2, legend “effect of a metric’s increase on fitness” and in the caption of Figure 2 which now reads: “Effects of metric’s increase on fitness and number of metrics per biological response category. Magenta, blue and gray fillings indicate metrics for which an increase leads to a negative, positive, or ambiguous effect on fitness, respectively.”

R1.C5: Figure 3: In the legend “summary of effects” would it not be more specific to use decrease or increase? Since otherwise it may be confused with effects on fitness, but I assume this is not what is meant as earlier in the manuscript it is indicated that it was not possible to ascertain for many metrics whether an effect is positive or negative for fitness.

Reply to R1.C5: We agree with the comment of the reviewer and changed “positive” and “negative” to “increase” and “decrease”. We did this in the legend and caption of former Figure 3, now Figure 4.

R1.C6: “Deviations of biological responses” first paragraph l. 8: the authors write that “by mathematical construction ... significant deviations can be found in the absence of significant directional changes”, and I completely agree. But maybe a short sentence explaining exactly why this is so would be useful for many readers not familiar in depth with the nature of response ratios and the problem that directional mean effect sizes can effectively “hide” both positive and negative effect sizes, cancelling each other out when the mean is taken. In fact, if this issue was introduced more clearly earlier in the manuscript (i.e. in the “Main” section), the brilliance of looking at the absolute value would be even clearer.

Reply to R1.C6: We thank the reviewer for their comment and appreciation of our approach. We edited the “Deviations of biological responses” paragraph to more clearly present why deviation changes can happen in the absence of a directional change, and why effect size cancelling in directional meta-analyses limit our understanding of climate change impacts. Starting from line 229 the text now reads: “When pooling different species and metrics, changes of opposite direction can cancel out, masking individually significant changes (Fig. 1). This is problematic as the deviation of any response from its reference state holds biological significance by altering the balance at the cellular, organism, or ecosystem scale. Deviation of responses requires thorough consideration and testing when evaluating climate change impacts and cannot be captured by meta-analyses based on relative effect size. For this reason, we converted relative effect size into absolute effect size ($|\ln RR|$) to calculate the average deviation in biological responses across studies. By mathematical construction,

all significant directional changes translate into significant deviations, but significant deviations can be found in the absence of significant directional change, because unlike relative effect sizes, absolute effect sizes do not cancel out when averaged (Fig. 1).” In addition, we also added a diagram (Fig. 1) showcasing the differences between directional changes and deviations and how antagonistic responses at the experiment level can cancel out when computing a mean directional change, while significant responses are revealed when computing mean deviation. This is done to introduce the issue earlier on in the manuscript. Please also see the changes in Reply to R1.C2.

R1.C7: “Conclusion” I. 4-6: the authors write that they “demonstrate that metrics commonly pooled in meta-analyses have predominantly ambiguous or context-dependent effects on fitness”. And while I certainly agree with the essence of the statement, it is unclear how exactly the authors “demonstrated” it. There is no detailed discussion or showing of data; the co-authors individually assessed the metrics, but there is no description of how the co-authors (= experts) argued for and against? There is no data or scores in Table S1, just the final classifications. Rather than “demonstrate” I would therefore use the word “argue”.

Reply to R1.C7: We changed “demonstrate” to “argue” as suggested and the lines 360-362 now read: “We argue that metrics commonly pooled in meta-analyses have predominantly ambiguous or context-dependent effects on fitness, which results in mean effect sizes that are difficult to interpret and that likely underestimate climate impacts.” Nevertheless, the assessment of metrics went through a formal expert elicitation approach which is described in lines 417 to 422: “The scores were given based on the expertise of five of the co-authors (KA, PD, MM, CC, FCM), who reviewed metrics and assigned them a direction independently (Supplementary Table 5). We adopted the most conservative approach, i.e., we only assigned a positive or negative direction to a metric if all five co-authors unanimously agreed on that direction. We classified all other metrics as having an ambiguous direction (Supplementary Table 5).” We added a supplementary table showing the individual scores from the co-authors (Supplementary Table 5).

R1.C8: “Literature search and data collection” second paragraph I. 4: It is specified exactly why it was necessary to restrict studies to those having done temperature and acidification in a full-factorial design, but I assume it has to do with how the effect sizes are interpreted, i.e. in terms of assessing whether they are additive/antagonistic? That would not be possible unless exactly the same studies are used, so maybe this could be clarified (if that is indeed the argument).

Reply to R1.C8: Yes, this is exactly why we restricted studies to those with a full-factorial design. We added a sentence starting in line 389 to explain this criteria: “This was done so that antagonistic, synergistic or additive effects of OA and OW could be evaluated.”

R1.C9: “Literature search and data collection” fourth paragraph I. 4: here it is explained that it is an INCREASE in a metric that is used as a basis to assess the effect on fitness, but this is not mentioned anywhere earlier, that I can see. But as I have commented on above, it gets confusing when it has not been mentioned.

Reply to R1.C9: We agree and edited the phrasing throughout the manuscript. Please see Reply to R1.C4 and Reply to R1.C5.

R1.C10: “Data analysis” first paragraph I. 3: Again, there is confusion about what “direction” means.

With “negative direction” I assume you mean “For metrics where an increase has a negative effect on fitness”, maybe better to specify more clearly?

Reply to R1.C10: Correct, this is what we meant. We now made the meaning of “negative direction” more explicit by rephrasing the sentence starting in line 471: “In the case of metrics for which an increase is detrimental to fitness (i.e., negative direction, Supplementary Table 2), the log of the inverse (i.e., control/treatment) was calculated, so that an increase would result in a negative effect size. This formula was also applied in the case of metrics of positive direction but with negative values, because an increase of a negative value corresponds to a negative outcome.”

R1.C11: “Data analysis” first paragraph l. 1-2 from bottom: The first things that strikes me is how weird it is that there were studies with zero variance. Not just small, but actually zero? Was it given as zero or was it not possible to assess from graphs? Next, “the value of 0.0001” was assigned”. Why this exact value? Why not 0.001 or 0.00001? While I do not know how many studies had this issue (maybe this could be mentioned?), and how big an impact it has for the overall results, I think it deserves a bit more explanation/justification.

Reply to R1.C11: Studies with null or extremely low variance (<0.0001) only happened for a few specific metrics, such as % mortality, % abnormal and % fertilization. In some studies, mortality was 100% across all treatments tested, leading to null variance. Similarly, some treatments led to consistent 0% morphological abnormality. Values in these two examples, indeed, corresponded to true null variance, and not some low variance values on graphs. However, it represented a very small proportion ($\sim 1\%$) of data points ($n=23$) belonging to 7 different studies for invertebrates and 1 data point for one study on fish whereas the total number of data points was 3162. The value of 0.0001 was chosen because it is the minimum threshold value accepted by the `rma()` function of the {metafor} R package to perform a meta-analysis. The impact of changing the variance value of these studies was checked during our sensitivity analyses, in particular by verifying that the weight given to these studies was within the range of the weight given to other studies within the same biological response category. This can be checked in the Supplementary Data 1, where the weight of each data point is given. Given the small number of concerned data points and their reasonable weight in the meta-analysis, we are confident that this choice does not have an impact on our results. We have now provided further explanations in the methods section regarding this point. Starting from line 486 the text now reads: “Experiments measuring survival, morphological abnormalities or fertilization success sometimes had null or extremely low within-study variance, e.g., as a result of all individuals surviving. Because the `rma()` function of the {metafor} package has a within-study variance threshold of 0.0001, we attributed the fixed value of 0.0001 to $n=24$ experiments (from a total of 3,162 experiments) for which variance fell under that threshold. We verified that this did not result in a disproportionate weight given to these data points by checking the weights attributed by models to these studies, as detailed in Supplementary Data 1.”

Reviewer #2 (Remarks to the Author):

This manuscript aims at re-evaluating the literature on both ocean warming, acidification and their combined effects. The general idea to deviate from the classic approach attributing a direction to metrics used to measure the biological impact is original and there is some interesting technical suggestions on how to focus on absolute deviation from a reference value. While the manuscript is

well-written, presented, and the data well analysed, there is a strong methodological flaw that compromises the selection of studies, the results, and the interpretation.

R2.C1: One critical factor explaining species sensitivity to temperature or the carbonate chemistry is local adaptation to the present natural variability. The variability is huge between regions but also at a given sampling site, especially on the coastal zone and in temperate regions. The importance of incorporating present natural variability has been recently illustrated by Vargas et al. (2017, 2022) for ocean acidification. While authors are citing these two papers to illustrate the importance of natural variability, they do not consider the main point that biological response is correlated to the deviation from the present range of natural variability in $p\text{CO}_2/\text{pH}$. Vargas et al. (2022) also makes the point that a majority of authors are using appropriate controls and treatments in their experimental studies. To resolve this issue, they propose an index that compare the high pH level to the extreme of the present range (instead of the “control” as defined by authors). A true OA and OW treatment should consider the present range of natural variability and chose scenarios that deviates from this range. Many authors are justifying their treatments using open ocean scenarios or neglect variability. When accounting for this, a lot of the apparent contradictions between studies disappeared. Another important point related to the natural variability is that for most metrics, there is not 1 control but a range of conditions (e.g. metabolism) that are within the present range of variability that changes without consequences on fitness.

To truly evaluate the effect of OA and OW (and their combination, which would require a stronger consideration of what is a relevant combination of scenarios), the treatments considered by the authors of the selected manuscript should be critically evaluated in the light of the present natural range of variability at the sampling site. A majority of authors were considering the wrong scenario for their “control” and “future treatment”.

Ignoring this leads to a mix of data on plastic response to natural variability with true response to OA and OW constitutes a strong methodological flaw and compromises the ability to perform a meaningful meta-analysis.

Reply to R2.C1: The suggestion of reviewer 2 to account for local environmental conditions was an important and helpful comment and we have taken considerable effort, including re-analysis, to address it.

Following this suggestion, we fully re-performed the part of our study which is sensitive to $p\text{CO}_2$ control values, i.e., the effect of climate driver intensity on biological responses, using local $p\text{CO}_2$ values instead of control $p\text{CO}_2$ values as given in publications. This involved a considerable amount of work to fully understand the approach developed in Vargas et al. (2022); review our 217 studies against the selection criteria defined in Vargas et al. (2022) and calculate $p\text{CO}_2$ exposure index for each experiment before re-running the analyses. We first reviewed each of our studies to evaluate whether or not they matched the criteria required to perform the approach developed in Vargas et al. (2022). This resulted in the elimination of a considerable number of studies, notably all studies on fish and mobile invertebrates. We kept all studies in our dataset that had been included in Vargas et al. (2022), and found an additional 37 studies that matched their selection criteria, leading to a total of 62 studies kept from our 217 original studies. We then ran two types of analyses. First, we tested the correlation between the magnitude of effect sizes (for both directional and deviational changes) and the treatment intensity level, using both the $p\text{CO}_2$ exposure index proposed in Vargas et al. (2022) and the $\Delta p\text{CO}_2$ using publication control values. We found a significant relation between treatment intensity

level and both directional and deviational responses using either $\Delta p\text{CO}_2$ approach. The significance and magnitude of this correlation was similar for both approaches. Second, we calculated the overall effect size per metric category and per $p\text{CO}_2$ intensity level, using either $\Delta p\text{CO}_2$ approach. While the number of experiments falling under each type of intensity level slightly changed depending on the $\Delta p\text{CO}_2$ approach, the combinations of biological response x climate driver for which significant responses were found remained very similar under both $\Delta p\text{CO}_2$ approaches. With both $\Delta p\text{CO}_2$ approaches, we found that extreme scenarios led to the highest number of significant relative responses across metric categories. In both $\Delta p\text{CO}_2$ approaches, we found that testing for deviational responses led to higher numbers of significant responses for lower intensity treatments (RCP 8.5 and RCP 6). As such, the main messages and novel findings of our study are confirmed using the approach developed in Vargas et al. (2022). Although we could not apply this approach to our entire dataset because of methodological constraints, the results arising from this subsample of studies clearly demonstrate the robustness of our results to variations in the choice of what is considered the most appropriate control value for climate stressors. This methodology can only be applied to organisms that are sessile or have low-vagility and when $p\text{CO}_2$ data from sampling sites are available and has not yet been extended to evaluate the effects of local temperature extremes. This would require a novel approach which is outside the scope of this study as explained in our methods (below).

A method section describing our additional analyses was added in lines 534-563 (inserted below), our results and discussion points are described in lines 209-223 (inserted below), and figures and tables showing results and statistical values are added (Supplementary Figure 1-4, Supplementary Table 12-13).

Line 534-563: *“Effect of upper environmental conditions.* We tested the effect of local upper environmental conditions, as a proxy for local variability, on the biological responses of organisms. We limited this analysis to OA following a detailed methodology developed to test the effects of local $p\text{CO}_2$ extremes on organisms’ responses to OA⁴⁶. This methodology can only be applied under a certain number of conditions, i.e., when studied organisms are sessile or have low-vagility and when $p\text{CO}_2$ data from sampling sites are available. Furthermore, it has not yet been extended to evaluate the effects of local temperature extremes. This would require a novel approach that takes into account other bioclimatic metrics such as diurnal temperature ranges, isothermality, temperature seasonality and range, microclimate as well as thermal acclimation capacity. In addition, many studies do not report the date of animal collection, the start date of experiments, and thermal conditions in the laboratory before commencement of experiments, which would be crucial information for such an approach. This adds to the difficulties associated with developing this approach, which is outside the scope of this study. We checked studies included in our meta-analysis against the selection criteria given in Vargas et al.⁴⁶. We retained studies for which sampling areas had been selected in Vargas et al.⁴⁶. We retained species selected in Vargas et al.⁴⁶ and included 24 additional sessile and gregarious or low-vagility benthic species (Supplementary Table 12). Out of the 217 studies used in our meta-analyses, 62 met all selection criteria, including 25 studies that were already included in Vargas et al.⁴⁶ and 37 additional studies (Supplementary Fig. 1).

Upper environmental conditions at the sampling sites of these 62 studies originated from global database and local buoys deployments, and were extracted from the supplementary information in Vargas et al.⁴⁶. We then calculated (1) a study-based $\Delta p\text{CO}_2$ and (2) a $\Delta p\text{CO}_2$ exposure index by calculating the difference between the $p\text{CO}_2$ treatment value and (1) the $p\text{CO}_2$ control value as given in studies, or (2) upper local environmental conditions, respectively.

We tested the relation between study-based $\Delta p\text{CO}_2$ and $\Delta p\text{CO}_2$ exposure index and the response of organisms using linear regression models. We attributed climate scenarios to each data point following the same procedure as described above (see *Climate scenario* section) but using the $\Delta p\text{CO}_2$ exposure index instead of the study-based $\Delta p\text{CO}_2$. Because studies that met the criteria necessary to calculate a $\Delta p\text{CO}_2$ exposure index were much fewer than our initial study pool, we performed tests at the biological response x intensity level regardless of sample sizes. Results from linear regressions are shown in Supplementary Figure 2 and directional and deviational responses by biological response and by intensity level using both $\Delta p\text{CO}_2$ approaches in Supplementary Figure 3 and 4, respectively. The model heterogeneity and residual heterogeneity associated with these models are shown in Supplementary Table 13.”

Line 209-223: “The intensity level of an experiment depends on the choice of its control value, which should account for the mean local environmental conditions but also for the variability and extreme conditions that organisms experienced during their development. However, $p\text{CO}_2$ control values used in studies are sometimes based on $p\text{CO}_2$ values for the open ocean, which can strongly differ from local coastal $p\text{CO}_2$ conditions⁴⁷⁻⁴⁸. For this reason, it has been suggested to measure intensity levels of experimental OA using a $\Delta p\text{CO}_2$ exposure index based on local $p\text{CO}_2$ upper conditions rather than on control values provided by studies⁴⁶. Applying this approach, we found a significant correlation between the $\Delta p\text{CO}_2$ exposure index and the magnitude of both directional and deviational responses, yet the data fit was similar to that based on $\Delta p\text{CO}_2$ as provided in studies (Supplementary Fig. 1-2). Similarly, responses of invertebrates to RCP 6, RCP 8.5, and extreme levels of OA were stable using either study-based or exposure index $\Delta p\text{CO}_2$, i.e., 75% and 79% of significant responses were shared using both approaches for directional and deviational effect sizes, respectively (Supplementary Fig. 3-4). While the exposure index approach is currently restricted to sessile organisms and $p\text{CO}_2$ treatments, adapting this methodology to accommodate the study of additional climate drivers and their combination, as well as mobile organisms, could provide further insights to elucidate drivers of organisms’ response to climate change.”

Reviewer #3 (Remarks to the Author):

The current manuscript provides a new approach to considering analyses of data sets on the impacts of climate warming and seawater acidification on marine species. In addition to the more traditional approach of providing a direction of change and assigning impact based on the change (i.e. positive or negative or unknown), this approach also considers deviations from control in response to some of the physical drivers of change in the ocean for invertebrates and fishes.

As a physiologist, I appreciated this approach as for many of the measurements the community has made in terms of responses to climate change, a response in a particular direction does not allow us to differentiate a positive response to a stressor to restore homeostasis vs. a negative long term response where performance is chronically impaired and reduced fitness would be a consequence. This is critical to accurately predict impacts of climate change. This deviations approach, combined with a directional approach is a good step in the right direction.

R3.C1: I appreciate the amount of work that goes into meta-analyses as extensive as this one. My question though is does it go far enough to provide significant changes in our predictive power on the biological responses to warming and acidification. There is still a fair amount of lumping of data sets, as the authors admit (Lines 319-329). With the focus on 4 factors, perhaps a couple of the most

important for understanding vulnerability (life stage) and experimental conditions that allow you to better understand whether the stress response is beneficial in the long term or not (acclimation time) are not considered. I am not sure how your data is coded but is this some additional analyses that could be run? Understanding which life stage is most vulnerable to deviations will allow for a better prediction of community effects since many fishes and inverts have bipartite life cycles and therefore their "communities" differ.

Reply to R3.C1: We agree that investigating the effects of acclimation time and life stage on responses would provide further insights into the predicted impacts of ocean warming (OW) and ocean acidification (OA). We have now extracted the number of acclimation days and the life stages of organisms for all our 217 studies and ran new analyses to test the effect of these two factors. For each of these two factors, we ran 12 different models, testing our two types of responses (deviational or directional) at the taxa (fish or invertebrate) x climate driver (OA or OW or OA+OW) level. We found that acclimation time only had a significant effect on a minority of taxa x climate driver combinations (1 out of 6 models for deviational effects, 3 out of 6 for directional effects) and this effect had a very low magnitude. By contrast, we found a strong effect of life stage on our results, with different trends when considering directional and deviational responses. While early life stages (embryos and larvae) typically had stronger negative responses than adults when considering directional responses, their deviational response tended to be smaller in magnitude than that of adults. This is likely due to the wider range of metrics tested on adults vs. embryos and larvae, and suggests that stronger responses of early life stages typically found in directional meta-analyses might not only be due to their greater vulnerability, but also to less counterbalancing effects from pooled metrics than for advanced life stages. These new analyses and findings are now described in our study in the method section (line 511-514, inserted below), in the main text (line 266-286, inserted below) and are shown in additional figures (Supplementary Fig. 5-8), with statistics shown in additional tables (Supplementary Table 4).

Line 511-514: "The influence of life stage (embryo, larvae, juveniles or adults) and acclimation time (number of days of acclimation, square-root transformed) on both relative and absolute effect sizes was investigated at the taxa x driver level. Model heterogeneity, residual heterogeneity, and associated p-values are provided in Supplementary Table 4."

Line 266-286: "**Effect of life stage and acclimation time.** Organisms' life stage (embryo, larvae, juvenile, or adult) had a significant effect on responses to climate drivers. In both fish and invertebrates, early life stages (embryo, larvae, juveniles) displayed more significant directional responses than adults (Supplementary Fig. 5-6, Supplementary Table 4). Early life stage invertebrates predominantly displayed significant decreases in responses (Supplementary Fig. 5) while early life stage fish displayed both significant increases and decreases. Under OW and OA + OW but not OA alone, biological responses of fish embryos were decreased and those of juveniles were increased. Biological responses of fish larvae were decreased under OA and increased under OW. For both invertebrates and fish, deviations of responses were significant and similar in magnitude across life stages and climate drivers, with the exception of embryos' responses that were lower in magnitude. Lower magnitude of deviations, but higher magnitude of directional response of embryos compared to more advanced life stages, could be due to the less ambiguous and less diverse metrics measured on embryos, typically related to survival and "normality" of developmental processes, leading to fewer counterbalancing effects when computing overall relative effect size. The higher sensitivity of early life stages to climate drivers has been found in some, but not all, previous meta-analyses and has been attributed to their lack of regulation and protection mechanisms to cope with environmental changes

(Sampaio et al.¹¹ vs. Cattano et al.¹⁴). Conversely, acclimation time had limited to no effect on directional and deviational responses of organisms (Supplementary Fig. 7-8). This is consistent with previous meta-analyses that did not find a clear effect of acclimation time on organisms' response⁷, and suggest that the influence of acclimation time is likely overshadowed by stronger drivers of responses at the meta-analytical scale, such as life stage, metric category, or intensity of climate driver level."

R3.C2: Lastly, there are a number of environmental physiology experts in this author list. I know the goal is to go beyond single species studies and extrapolate to community level impacts but I would have appreciated a physiologist view of why deviations from controls (away from homeostatic set points) in either direction constrain energy budgets and can lead to fitness consequences if homeostasis is not restored.

Reply to R3.C2: We thank the reviewer for this comment. We agree that more framing and explanation was useful to readers. We dedicated a paragraph in the manuscript to address it. From line 289 to line 298 the text now reads: "The relevance of examining deviational effects of climate drivers is linked to characteristics of biological processes from the cellular to the ecosystem level. Over evolutionary time scales, organisms have adjusted their metabolic machinery to achieve physiological homeostasis at the lowest metabolic cost possible within the range of conditions of their local environment⁴⁹. Any deviation from an optimal setpoint of homeostasis, whether originating from a metric increase or decrease, is energetically costly as it induces metabolic regulation and, in some cases, compensatory responses⁵⁰. If abiotic conditions vary within the evolutionarily experienced maxima and minima, physiological regulation will ensure homeostasis, yet regulatory metabolic costs will usually rise with increasing deviation from the setpoint⁵¹. As such, deviation in physiological responses might provide a valuable indicator of the level of stress that organisms are experiencing."

REVIEWERS' COMMENTS

Reviewer #1 (Remarks to the Author):

The authors have addressed my comments, and I think clarity has been much improved through text edits and the new fig 1. I have no further comments other than to repeat that I think this is an important contribution.

Reviewer #3 (Remarks to the Author):

I really appreciate the amount of effort and commitment went into addressing Reviewer #2 and my comments. We both brought up concerns that required additional analyses and the authors did a fantastic job at exploring their models and additional factors very thoughtfully. These additional analyses, some of which brought up new ideas (early life history stages) and some of which confirmed existing findings (control vs. natural variability of PCO₂) are excellent additions to this already thorough analysis. Well done! This will be an important paper for the field. I have no additional concerns.